

# Case study of spatial and temporal variability of snow cover, grain size, albedo and radiative forcing in the Sierra Nevada and Rocky Mountain snowpack derived from imaging spectroscopy

F. C. Seidel[1], K. Rittger[2], S. M. Skiles[1], and T. H. Painter[1]

[1]Jet Propulsion Laboratory / California Institute of Technology, Pasadena CA, USA
[2]National Snow and Ice Data Center, University of Colorado, Boulder CO, USA

*Correspondence to:* F. C. Seidel (Felix.C.Seidel@gmail.com)

**Abstract.** Quantifying the spatial distribution and temporal change in mountain snow cover, microphysical and optical properties is important to improve our understanding of the local energy balance and the related snowmelt and hydrological processes. In this paper, we analyze changes of snow cover, optical-equivalent snow grain size, snow albedo, and radiative forcing by Light Absorbing Impurities in Snow and Ice (LAISI) with respect to terrain elevation and aspect at multiple dates during

the snowmelt period. These snow properties are derived from Airborne Visible/Infrared Imaging Spectrometer (AVIRIS) data from 2009 of the maritime snowpack in California's Sierra Nevada and from 2011 of the continental snowpack in Colorado's Rocky Mountains, USA.

Our results show a linearly decreasing snow cover during the ablation season in the Rocky Mountains and a snowfall driven change in snow cover in the Sierra Nevada. At the same time, the snow grain size is increasing primarily at higher elevations

and north facing slopes from 200 microns to 800 microns on average. We find that intense snowmelt renders the mean grain size almost invariant with respect to elevation and aspect. Our results confirm the inverse relationship between snow albedo and grain size, as well as between snow albedo and radiative forcing by LAISI. At both study sites, the mean snow albedo value decreases from approximately 0.7 to 0.5. The mean snow grain size increased from approximately 150 to 650 microns. The mean radiative forcing increases from $20 \, \mathrm{W \, m^{-2}}$ up to $200 \, \mathrm{W \, m^{-2}}$ during the ablation period. The variability of snow albedo

and grain size decreases in general with the progression of the ablation period. The spatial variability of the snow albedo and grain size decreases through the melt season while the spatial variability of radiative forcing remains constant.

## 1 Introduction

Snow is important to the hydro- and biosphere, to mountain and arctic regional climates, as well as to water resources management as a natural water storage. Snow influences the Earth's energy balance through its strong albedo feedback. Changes

to the snow cover extent, snow albedo and snow water equivalent, therefore have direct impacts on the environment, water availability and economics.

Snowmelt in mid-latitude mountains is largely dominated by the absorbed solar radiation (Marks and Dozier, 1992; Oerlemans, 2000; Painter et al., 2007a). The net shortwave radiation is a function of the solar zenith, cloud cover, surface albedo, as





well as terrain elevation, slope and aspect. The solar irradiance at the surface is highly variable in mountainous areas mainly due to the complex terrain, cloud cover, and vegetation above the snow surface (Link and Marks, 1999; Garen and Marks, 2005). Besides snowmelt, snowfall impacts snowpack properties very rapidly and thus, it is important to capture temporal changes in addition to spatial patterns.

Snow microstructure is a key indicator of physical processes occurring in a snowpack. Snow grain size is generally used as a proxy for the microstructure due to the lack of a direct measurement technique. Throughout this paper, we refer to the optical-equivalent snow grain radius ($r$), or simply snow grain size for brevity. It can be estimated with handheld spectroscopy instruments (Painter et al., 2007b; Skiles, 2014) as well as airborne imaging spectroscopy (Nolin and Dozier, 2000; Painter et al., 2003, 2013). The former can be used to resolve the vertical distribution of snow grain sizes within a snowpack, and the

latter is very useful to resolve the spatial and temporal distribution of not only snow grain size but also albedo and radiative forcing on scales useful for snow hydrology. It should be noted that $r$ should not be directly compared with the snow grain size derived using other measurement techniques, such as for example the traditional visually estimated snow grain size or the specific surface area (SSA). Gallet et al. (2009) and Leppänen et al. (2015) suggest a conversion of $r = 3/(\rho_{ice}SSA)$, where $\rho_{ice}$ is the density of ice ($917 \text{ kg m}^{-2}$). An optical-equivalent snow grain radius of $100 \ \mu\text{m}$ and $650 \ \mu\text{m}$ corresponds therefore

to a SSA of $33 \text{ m}^2 \text{ kg}^{-1}$ and $5 \text{ m}^2 \text{ kg}^{-1}$, respectively.

Snow albedo is mainly changed by snowfall, metamorphism, and light absorbing impurities in snow and ice (LAISI). While fresh snow with small grains has a high albedo value, aged snow has a lower albedo value in conjunction with snow grain size growth and change in optical path length. LAISI, such as black carbon, dust, and tree litter have the most profound impacts on snow albedo (Warren and Wiscombe, 1980; Qian et al., 2015).

This 'darkening' of snow increases the absorption of solar irradiation, which reduces the cold content of the snowpack or enhances snowmelt if its temperature is at 0°C. The related consequences of LAISI to snow hydrological and climatological processes became more evident by the work of Ramanathan et al. (2007); Painter et al. (2007a); Qian et al. (2009); Flanner et al. (2009); Painter et al. (2010, 2012b) and others, and novel remote sensing algorithms were developed by Painter et al. (2012a, 2013) to retrieve the radiative forcing by LAISI.

The combination of Airborne/Visible Infrared Imaging Spectrometer (AVIRIS) data (Sect. 2.4) with our algorithms (Sect. 3) allow us to address the temporal variation and spatial distribution of snow properties in two partial hydrological catchment areas located in the Rocky Mountains, Colorado and the Sierra Nevada, California, USA (Sect. 2.2) as presented in Sect. 4 and discussed in Sect. 5.

## 2   California and Colorado study area and data description

In the Western United States, seasonal snowmelt provides more than 70% of water supply. The Sierra Nevada in California and the Rocky Mountains in Colorado are examples of two mountain ranges where reservoirs store snowmelt for use in late spring and throughout the summer and early fall when precipitation is less frequent. As such, these regions are of great interest to snow research and water management and we analyze data in each.





## 2.1 Terrain

We use the United States Geological Survey National Elevation Dataset (NED) with one arc second spatial resolution (roughly 30 m). A hill shaded version is used as background to Fig. 1. The Sierra Nevada study region covers a larger distribution of elevations with a vertical range of 2400 m while the Colorado study area has a vertical range of only 1200 m (Fig. 2). The

Colorado study area reaches higher elevations with a maximum at 4111 m, while the maximum elevation in the California study area is only 3533 m. The mean elevation is over 1000 m higher in the Colorado study area (Table 1). The Kaweah and Kings River basins in Sierra Nevada study area drain to the West as seen in the distribution of aspects with a peak around $270°$ in Fig. 2. The Uncompahgre River basin drains to the North hence the lower percentage of $180°$ slopes. A larger proportion of the Uncompahgre slopes face East and West draining toward the center of the basin explaining the larger percentage of the

basin with aspects of $90°$ and $270°$.

## 2.2 Snow hydrology

In the Sierra Nevada of California, the study area covers part of the Kings and Kaweah river basins. In the Rocky Mountains of Colorado, the study area covers the Senator Beck drainage, which is part of the Uncompahgre watershed (Skiles et al., 2012; Painter et al., 2012b, 2013). The Kings and Kaweah have maritime snowpacks (typically warmer and deeper) while

the Uncompahgre has a continental mountain or Alpine snowpack (typically colder and more shallow) (Sturm et al., 1995), allowing us to test our algorithms in different snow regimes.

We use snow pillow data of snow water equivalent (SWE) at high and low elevation as a proxy for snow accumulation or snowmelt as shown in Fig. 3. We choose the Big Meadow (36.72N, 118.84W, 2300 m., Kings) and the Farewell Gap (36.41N 118.58W, 2900 m., Kaweah) stations to represent the California study area. For Colorado, we choose the Idarado (37.93N,

107.68W, 3000 m., Uncompahgre) and the Red Mountain Pass (37.90N, 107.72W, 3400 m., Uncompahgre) SNOTEL stations. Based on those SWE data, 2009 was an average year in the Sierra Nevada, and 2011 was an above average year in the Rocky Mountains. Peak SWE and length of the ablation season were fairly similar at the two high elevations stations in each area, but separated by a month. At Farewell Gap SWE peaked ca. 80 cm on 17 April 2009 and snow melt out occurred on 19 May 2009. At Red Mountain Pass SWE peaked at ca. 85 cm on 25 May 2011 and melt out occurred on 20 June 2011.

This is not to say that the two study areas are similar, in additional to regional and interannual variability, there was some distinct differences in melt patterns. In Kings/Kaweah multiple snowfall events refreshed the snow surface over the period of flight, delaying the main snowmelt pulse until just prior to snow melt out. In the Uncompahgre melt initiated just after peak SWE and steadily increased over the ablation period, with the exception of a minor snowfall event at the end of May. While the measurement sites may not be wholly representative of the two study areas, they are helpful in interpreting some of the spatial

and temporal variability exhibited in the acquisitions, discussed in Sect. 4.





### 2.3 Dust concentrations in snow

Senator Beck Basin is a research catchment established to monitor the hydrological impacts of regularly observed episodic events of dust deposition on snow in the Upper Colorado River Basin (Painter et al., 2012b). Dust is particular effective at altering snow energy balance in this region because the majority of mass is deposited in the spring coinciding with snowmelt
onset (80% of events occur March-May), and additionally, dust is inefficiently scavenged by melt water and layers coalesce at the surface as snow melts compounding albedo decay. Over the ten year record (2005-2015) end year dust concentrations have ranged from $0.83$ to $4.83 \, \mathrm{mg \, g^{-1}}$, and exhibit a bimodal distribution, with about $1 \, \mathrm{mg \, g^{-1}}$ in lesser dust years and over $4 \, \mathrm{mg \, g^{-1}}$ in extreme dust years (Skiles et al., 2015). There were nine observed dust events in 2011 after March 1st and the end of year dust concentration was $1.3 \, \mathrm{mg \, g^{-1}}$, making it a lesser dust year. Black carbon has only been monitored across the
2013 ablation season, and concentrations were so low ($<1$ to $20 \, \mathrm{ng \, g^{-1}}$) that it was concluded that its contribution to radiative forcing in this region is negligible in the presence of such heavy dust loading (Skiles, 2014).

Unfortunately, there is not a similar data set for any basin in the Sierra Nevada, but given the predominant wind patterns and location of dust emission sources in the western US, we would not expect as great of dust loading there. There was a study conducted in the eastern Sierra Nevada in 2009 which found surface black carbon concentrations of $<1$ to $75 \, \mathrm{ng \, g^{-1}}$ and
dust concentrations of $1$ to $44 \, \mu\mathrm{g \, g^{-1}}$ over the ablation season, despite the lower concentrations sampled in the Sierra Nevada relative to the Rocky Mountains, this study also concluded that dust dominated the radiative forcing (Sterle et al., 2013).

### 2.4 Imaging spectroscopy data

Results provided in this paper were derived from georeferenced at-sensor radiances acquired by the NASA/JPL Airborne Visible/Infrared Imaging Spectrometer (AVIRIS) flown on NASA's ER-2 high altitude aircraft and on a Twin Otter aircraft.
Table 2 shows specific information for each flight line including aircraft altitude and heading, terrain elevation, pixel size, solar geometry, as well as retrieved columnar water vapor. The pixel size is directly related to the terrain height and the aircraft altitude, which varied significantly between acquisition dates. We therefore resampled the AVIRIS data to the common coarsest resolutions of $14.5 \, \mathrm{m}$ and $13.8 \, \mathrm{m}$ for California and Colorado, respectively. For each acquisition date, the images are mosaicked and cropped to their common area ($252 \, \mathrm{km^2}$ in California and $54.7 \, \mathrm{km^2}$ in Colorado). This enables inter-comparisons between
acquisition dates at the pixel-level. Figure 1 shows the outline of each flight line overlaid on the terrain for each study area (intersection of all flight dates) as a hatched line. The 27 February 2009 and 13 May 2011 lines for California and Colorado respectively lie directly on the study area because they covered the smallest area.

## 3   Remote sensing retrievals of quantitative snow properties

The quantitative snow properties shown and analyzed in this paper are extracted from imaging spectrometer data using retrievals
based on a legacy of algorithms by e.g. Nolin and Dozier (2000); Painter et al. (2001, 2003); Painter and Dozier (2004); Green





et al. (2006); Dozier et al. (2009). Our retrieval algorithms are described in details in Painter et al. (2013) and summarized in this section for reference.

### 3.1 Directional reflectance

Reflected and scattered solar radiation are observed by the airborne spectrometer and transformed into calibrated radiance for
each wavelength band $\lambda_i$, based on band-wise calibration.

The retrieval of surface properties requires that we convert the measured radiance first to reflectance at the sensor level $R_{sensor}^{obs}$ and then to surface directional reflectance by compensation of atmospheric scattering and absorption. We use the Atmospheric / Topographic Correction for Airborne Imagery (ATCOR-4) (Richter and Schläpfer, 2014) software to model the reflectance of the atmosphere at the sensor level $R_{atm}^{mdl}$, the total transmittance $T^{\updownarrow,mdl}$, and the spherical albedo at the surface
level $S_{sfc}^{mdl}$ to derive the spectral Hemispherical-Directional Reflectance Factor at the surface level from $R_{sensor}^{obs}$ (Schaepman-Strub et al., 2006):

$$HDRF_{sfc}^{obs} = \frac{R_{sensor}^{obs} - R_{atm}^{mdl}}{T^{\updownarrow,mdl} + S_{sfc}^{mdl}\left(R_{sensor}^{obs} - R_{atm}^{mdl}\right)}. \tag{1}$$

Note that the rugged terrain in both study areas (see Sect. 2) requires us to account for the local solar and viewing angles at each terrain facet. Local zenith angles are derived by:

$$15 \quad \theta'_{[\odot,v]} = \arccos\left(\cos\theta_{[\odot,v]} \cdot \cos slp + \sin\theta_{[\odot,v]} \cdot \sin slp \cdot \cos\varphi'\right), \tag{2}$$

where $\varphi' = \varphi - asp$, $slp$ and $asp$ denote the terrain's slope and aspect, respectively. All angles are defined in degrees. Although the following geometries are locally defined with respect to the terrain, we omit its notation for readability. Note that we mask pixels affected by terrain shadows in our retrievals, except for the snow cover determination.

### 3.2 Snow cover

A byproduct derived from $HDRF_{sfc}^{obs}$ is the snow cover map. We apply the normalized difference snow index (NDSI):

$$NDSI \equiv \frac{HDRF_{sfc}^{obs}(\lambda_{VIS}) - HDRF_{sfc}^{obs}(\lambda_{SWIR})}{HDRF_{sfc}^{obs}(\lambda_{VIS}) + HDRF_{sfc}^{obs}(\lambda_{SWIR})}, \tag{3}$$

where $\lambda_{VIS} \approx 0.6$ $\mu$m and $\lambda_{SWIR} \approx 1.5$ $\mu$m (Dozier and Marks, 1987; Dozier, 1989; Hall et al., 2002). To mask partially covered snow pixels, we empirically determined NDSI thresholds of 0.90 for the Sierra Nevada and 0.93 for the Rocky Mountains by visual interpretation with the corresponding true-color images. The high spatial resolution of the AVIRIS imagery
(see Table 2) allows the use of this simple technique without rejecting too many pixels. The multiplication of the number of remaining snow pixels by the pixel size yields the approximate (low-biased) snow covered area. We sum the snow covered area and normalize it with the study area to report the approximate percent snow cover.





### 3.3 Snow grain size

The optical-equivalent snow grain size, reported here as radius $r$, is an important snow property and prerequisite for the snow albedo retrieval.

Based on Nolin and Dozier (2000), we derive the snow grain size by searching for the best fit between the observed
$HDRF_{snow}^{obs}$ and modeled $HDRF_{snow}^{mdl}$ snow spectra with the minimum sum of absolute differences:

$$r : \left[ \min \sum_{i=1.17\ \mu m}^{1.27\ \mu m} \left| HDRF_{snow}^{mdl} (\theta_{\odot}, \theta_v, \varphi; r; \lambda_i) - HDRF_{snow}^{obs} (\theta_{\odot}, \theta_v, \varphi; r; \lambda_i) \right| \right]. \tag{4}$$

We assume to observe a single layer with a homogeneous snow grain size. Using the ice absorption feature centered around $1.27\ \mu m$ allows us to probe the uppermost layer in the snowpack with a depth in the order of $1\ cm$ (Nolin and Dozier, 2000; Gallet et al., 2009). $HDRF_{snow}^{mdl}$ is modeled using the radiative transfer code DISORT (Stamnes et al., 1988; Painter et al., 2003). The two spectra are matched at the corresponding solar and viewing geometry for each pixel in the terrain. The fit is performed at AVIRIS band numbers 86 to 99, which correspond to the wavelength interval $\lambda = [1.17\ \mu m, 1.27\ \mu m]$. The use of these bands reduces the bias by liquid water on light absorption by frozen water. This simple, yet powerful approach allows more robust retrievals of the optical snow grain size under snowmelt conditions (Painter et al., 2013).

### 3.4 Snow albedo

Albedo is a key parameter to the Earth's energy balance. The snow albedo feedback mechanism has a strong impact on climate on all temporal and spatial scales. In particular, snow albedo is a dominant driver of snowmelt, ecohydrological processes, as well as water supplies.

We derive the spectral snow albedo $\alpha_{snow}^{obs}$ from $HDRF_{snow}^{obs}$ by applying the anisotropy factor, which is given by the ratio between the modeled spectral albedo and the spectral $HDRF_{snow}^{mdl}$:

$$\alpha_{snow}^{obs} (r, \lambda) = \frac{HDRF_{snow}^{obs} (\theta_{\odot}, \theta_v, \varphi; r; \lambda)\, \alpha_{snow}^{mdl} (r, \lambda)}{HDRF_{snow}^{mdl} (\theta_{\odot}, \theta_v, \varphi; r; \lambda)} \tag{5}$$

The integration of Eq. (5) normalized by the topographically-modulated spectral irradiance at the surface level $E_{sfc}^{mdl} (\lambda, \theta_{\odot})$ derives the broadband (spectrally integrated) albedo as a function of the snow grain size $r$:

$$\alpha_{snow}^{obs} (r) = \frac{\int_{\lambda=0.36\ \mu m}^{2.5\ \mu m} E_{sfc}^{mdl} (\lambda, \theta_{\odot})\, \alpha_{snow}^{obs} (r, \lambda)\, \Delta\lambda}{\int_{\lambda=0.36\ \mu m}^{2.5\ \mu m} E_{sfc}^{mdl} (\lambda, \theta_{\odot})\, \Delta\lambda}. \tag{6}$$

$E_{sfc}^{mdl} (\lambda, \theta_{\odot})$ is provided as an output from ATCOR-4 (Painter et al., 2013; Richter and Schläpfer, 2014). Note that albedo values must range between 0 and 1, while HDRF can range up to 1.4 or even more.

### 3.5 Snow radiative forcing by light absorbing impurities

We calculate the radiative forcing by LAISI (RF) based on the difference between the observed spectral albedo and a modeled spectral clean snow albedo of the same grain radius (Painter et al., 2013). This reduction of spectral albedo of snow by





impurities is wavelength-wise multiplied by the total irradiance resulting in the additional radiative flux into the snow pack:

$$RF_{snow} = \int_{\lambda=0.36\ \mu m}^{1.08\ \mu m} E_{sfc}^{mdl}\left(\lambda, \theta_\odot\right) \left[\alpha_{snow}^{mdl}\left(r, \lambda\right) - \alpha_{snow}^{obs}\left(r, \lambda\right) c\left(r, \lambda_c\right)\right] \Delta\lambda, \tag{7}$$

where $c\left(r, \lambda_c\right) = \alpha_{snow}^{mdl}\left(r, \lambda_c\right) / \alpha_{snow}^{obs}\left(r, \lambda_c\right)$ denotes a scalar to adjust the modeled spectra to the measurements for a potential bias by terrain modeling imperfections. $c$ is derived at $\lambda_c = 1.08\ \mu m$ where the influence of dust is relatively small (Warren and Wiscombe, 1980).

### 3.6 Estimates of the retrieval uncertainties

This study addresses mainly the qualitative nature of the snow property distribution in mountainous terrain. However, we have estimated the retrieval uncertainties based on the AVIRIS sensor performance (noise-equivalent changes in radiance) and the retrieval method. The results were published by Painter et al. (2013), and we indicate, therefore, only the main points here. The uncertainties of snow grain size, albedo, and radiative forcing were estimated on the order of $\pm 25\ \mu m$, $\pm 0.0001$ (unitless), and $\pm 4\ \mathrm{W\ m^{-2}}$, respectively. The modeled snow spectral reflectance is based on the assumption of spherical grains due to the use of the Mie theory (Mie, 1908). This assumption becomes more valid with grain growth associated with evolving metamorphism and rounding of the snow crystals. Nevertheless, strong snowmelt with percolation of meltwater is assumed to introduce yet unknown bias to the snow grain size retrieval. More detailed studies of optical properties of melting snow would be required, such as Gallet et al. (2014) for example. A further discussion on retrieval uncertainties is given in Sect. 4.5 based on the actual retrieval results.

## 4   Results

Figs. 4 to 6 show maps of snow grain size, albedo, and radiative forcing in the Sierra Nevada (upper panel) and Rocky Mountains (lower panel) derived using algorithms described in Sect. 3 and Painter et al. (2013). The three dates represent an early, mid, and late observation during the melt period (see Fig. 3). These results are further analyzed in Sect. 4.1 regarding the temporal changes and in Sects. 4.2 to 4.4 regarding the spatial distribution and correlation with respect to elevation and aspect of the terrain.

### 4.1   Temporal changes of regional optical snow properties

In general, the snow line is significantly lower in the Sierra Nevada study area as compared to the Rocky Mountains because of the maritime snowpack. However, inter-annual variability also modulates the snow covered area. We calculate the snow covered area of the study area and normalize it by the basin area to serve as a proxy for accumulation and melt (Sect. 3.2). Figure 7 shows that the sequence of images in Kings/Kaweah captures the transition from an accumulating to melting snowpack, while it is mainly the melt season in the Uncompahgre (refer also to Fig. 3). We can observe a difference between the two areas in the rate of snow cover change with respect to time, which is an indicator for the intensity of snowmelt. This is likely related to




both the more frequent snowfall in the Kings/Kaweah in April and difference in solar irradiance between the March-April and May-June time periods.

The snowfall event between the first two airborne acquisitions in the Sierra Nevada also decreased the mean snow grain size from approximately 330 to 180 $\mu$m, and then varied little between the second and third acquisitions, after which it increased
to 650 $\mu$m by the 4th and 5th acquisitions a month later. From the snow measurement sites we know that the transition to a melting snowpack likely occurred between these two acquisitions (see Fig. 3), while this tends to happen later at the higher elevations, such a large increase in grain size across the study area is not totally unexpected. See Sect. 5 for a discussion on the inverse relationship between snow grain size and elevation under intense snowmelt conditions.

Patterns in mean broadband albedo are similar to those exhibited for grain size. The Sierra Nevada study site remained
constant at around 0.7 due to late season snowfall until April and then decreased to 0.55 by the last two acquisitions in May, when the snowpack was melting. In the Rocky Mountain study area the mean snow albedo decreased from 0.75 in May to 0.5 in late June. In both areas the largest decreases were exhibited in May, when both sites were likely undergoing the transition from the retention of cold content (below 0°C) to isothermal (at 0°C) and melting.

The time-series of radiative forcing are very similar for the two study areas, except that the values in the Rocky Mountains
are roughly 20 W m$^{-2}$ larger. This is consistent with the high dust concentrations in the Rocky Mountains (Skiles et al., 2012; Painter et al., 2012b) in combination with the increased solar irradiance later in the season.

## 4.2 Spatial distribution of snow grain size

Snow grain sizes at top of the snow pack are affected by snowfall, metamorphism and melt processes. We would expect to find smaller snow grains at higher elevation and north-oriented slopes. With time, we also expect snow surface grain size to grow
at varying rates, but decrease abruptly with the addition of snowfall. The analysis of our retrievals using Eq. (4) is consistent with this common theory. The two-dimensional histograms in Figs. 8 and 9 show the relation of grain size with elevation and with terrain aspect.

For both study areas we find for the early ablation period, that the snow grain size at high elevations is less than 200 $\mu$m and 500 – 800 $\mu$m at low elevations. The snow grain size is inversely correlated with elevation in the May data of the Rocky
Mountains. Snow grain size gradually increases over time at higher elevation as shown in the left panels of Figs. 8 and 9. Late in the ablation period the snow grain size is nearly constant with elevation, ranging from 500 $\mu$m to 700 $\mu$m and becomes positively, but less sensitively, correlated with elevation. Potential reasons for this remarkable change over time are discussed in Sect. 5. The snow grain size distribution with respect to the terrain orientation (aspect) is shown in the right panels of Figs. 8 and 9. As expected, we retrieve small snow grain sizes, on the order of about 100 $\mu$m on north facing slopes, versus 400 $\mu$m
or more on south-east slopes. On 5 March 2009, just after a snow event, the snow grain sizes are significantly smaller with a small variability, especially at higher elevations. More pixels at lower elevations were covered with snow (see Fig. 8).





### 4.3 Spatial distribution of snow albedo

The reflectivity of snow determines how much solar irradiance is absorbed, which provides the most energy for reduction of cold content and melting. Imaging spectroscopy data allow us to derive the HDRF (Eq. 1), and from that metric the spectrally integrated albedo with a snow reflectance model (Eq. 6). The broadband albedo is directly relevant to the energy balance in a
snowpack and, therefore, a main product of our retrievals (see Sect. 3).

   Snow albedo is correlated with elevation and terrain aspect early in the ablation period at both study areas (see Figs. 10 through 13). Our retrievals in the early ablation period show albedo values of 0.6 to 0.8 at higher elevations, and 0.5 to 0.6 at lower elevations for both study areas. A rather large spatial variance is observed in the February, March and April data in the Sierra Nevada Fig. (10) with albedo varying from 0.5 to almost 0.9 within less than a kilometer. In each case the higher albedo
values are found on north and steep west facing slopes. Later in the ablation period, albedo values decreased to approximately 0.5 and 0.4 at high and low elevations, respectively. In general the albedo increase with elevation is more distinct at the Sierra Nevada study area than in the Rocky Mountain study area. A strong dependence of snow albedo with terrain slope is found in February and March in the Sierra Nevada. North facing slopes show values ranging between 0.8 and 0.9, whereas south facing slopes are clearly darker with approximately 0.6 albedo values (Figs. 10 and 12). We found similar, although less distinct,
dependencies in the Rocky Mountains (Figs. 11 and 13). This is likely because northern slopes experience larger relative changes in irradiance during spring due to a decrease in solar zenith angle and the related reduction of terrain shadows. The Rocky Mountain study area observations were started in May, so the terrain effects are less prominent than those of the Sierra Nevada study area, which were started in February. During the late ablation period the differences in snow albedo between north and south facing slopes was less than 0.1.
In both time-series the spatial variance in the retrieved albedo corresponds with the spatial variance in retrieved snow grain sizes (see Fig. 7). For example, the retrievals on 5 March 2009, just after the snowfall event, show a increased variances in both albedo and grain size (see Fig. 12).

### 4.4 Spatial distribution of radiative forcing

We find the radiative forcing increasing during ablation from almost zero to spatial mean values of 170 W m$^{-2}$ in the Sierra
Nevada and 190 W m$^{-2}$ in the Rocky Mountains (Figs. 7, 14, and 15). The standard deviations are in the order of 60 W m$^{-2}$ in both study areas across the full period. The Colorado data were acquired later in spring than the California data, and thus with more solar irradiance. A direct quantitative comparison of radiative forcing between the two datasets is therefore difficult. The higher elevations in Colorado contribute to a later onset of snowmelt as compared to the Sierra Nevada, which leads to a later exposure of LAISI at the surface of the snowpack. We should bear in mind that each year is different in terms of dust
deposition and exposure at the surface (Painter et al., 2012b). Nevertheless a qualitative comparison with changes in snow water equivalent (Fig. 3), as an indicator for snow accumulation and snowmelt, shows a relationship of longer snowmelt periods with an increase in radiative forcing. This is likely due to higher impurity content in the Rocky Mountains, and with a longer time period over which impurities can be deposited and then accumulated at the surface. In 2011 only three of the observed dust





events occurred between the first and last flights in the Rocky Mountains, but all of the dust deposited over the full season would have surfaced as snow started to melt likely contributing to the higher radiative forcings in this area.

In California, we found almost no dependence of radiative forcing on elevation (Fig. 14). In contrast, the radiative forcing retrieved in Colorado is clearly correlated with respect to elevation with a distinct trend of larger radiative forcing at lower

elevations (Fig. 15). Column 2 of Figs. 14 and 15 shows the radiative forcing with respect to the terrain aspect. In the northern hemisphere, south facing slopes experience stronger irradiance and decreased snow albedo values (Sect. 4.3). The combination of both of these factors, in addition to LAISI deposition, led to increased radiative forcing. For example, in the Sierra Nevada, the radiative forcing is close to zero until May at northern slopes and up to about $150$ W m$^{-2}$ at south-south-eastern aspects. In May, we retrieve up to $150$ W m$^{-2}$ at northern slopes and about $200$ W m$^{-2}$ at southern oriented terrain slopes (Fig. 14).

The analysis of the Rocky Mountain study area shows a similar trend over time, but with radiative forcing values $50$ W m$^{-2}$ to $100$ W m$^{-2}$ larger than in the Sierra Nevada (Fig. 15). We also find that the dependence of terrain aspect angle with radiative forcing decreases until the summer solstice due to the decreasing solar zenith angle and the corresponding irradiance increase on steep slopes in mountainous terrain. Our results show radiative forcing differences between north and south facing slopes being smoothed out over time. An exception is found in late June in Colorado with significantly larger radiative forcing on

south-east slopes of up to $300$ W m$^{-2}$. This is presumably due to the earlier melt out of south to south-west slopes and may also be due to dust transport effects.

## 4.5  Uncertainty and error analysis

An indirect estimation of the data quality and retrieval robustness can be performed by an analysis of the variance on timescales shorter than the natural variability. For example, the last two Sierra Nevada datasets are a single day apart and the snow products

were retrieved with almost identical values. The (relative) differences between 8 May and 9 May are -2.8km$^2$(-1%) snow cover, -0.01(-2%) albedo, +35 $\mu$m(+5%) grain size, and +3 W m$^{-2}$(+2%) radiative forcing (see Fig. 7). These are all plausible values, which can be at least partly attributed to physical processes as opposed to substantial errors.

An additional source of errors is the insufficiently spatially-resolved digital elevation model. The elevation model used in this study was interpolated (cubic convolution) from 30 m (Sect. 2.1) to the actual pixel size of each AVIRIS flight line, which

is on the order of 2 to 15 m (Tab. 2). Terrain effects are also a source of errors, especially if shadow areas in steep terrain are not correctly accounted for. As a mitigation, we have masked retrievals in shadow areas, except for the snow cover calculations. Up to 10% of the snow pixels are affected by terrain shadows in the February data due to large solar zenith angles. The shadows increase the variance, or noise, in the retrieval products.

## 5  Inverse relationship of grain size with elevation of melting snow

As grain sizes in a dry snowpack are primarily driven by metamorphism, itself driven by temperature and vapor pressure gradients, we expect increased snow grain sizes at lower elevations because of warmer temperature and higher positive energy balance. This is in agreement with our observations shown in Figs. 8 and 9 of the initial ablation period in both study areas.



However, we find the snow grain sizes at higher elevations increase significantly late in the ablation season, while they remain mostly unchanged at lower elevations. In the Sierra Nevada this is apparent in the May acquisitions relative to the April acquisition. In the Rocky Mountains this is apparent in the June acquisitions. Later in the ablation season, with increased solar irradiance, the snow grain size distribution becomes independent of the terrain aspect.

We suggest that there are two processes that account for these observations. First, that changes in grain size are slowest when the snowpack is both cold and when it is fully isothermal, and most rapid during the transition between these two phases (Taillandier et al., 2007). When the grain growth is slow at lower elevations, but rapid at higher elevations, this indicates that lower elevations have become isothermal while the upper elevations are still undergoing that transition. Grain growth can be rapid in cold snowpacks, but this tends to occur near the ground where the temperature gradient between the ground and snow

is greatest, and not at the surface. When liquid water is preset snow grains tend to appear visually larger because they form clusters but optically the absorbing path length seems to change very little. This same slowing of grain growth rates after the isothermal transition was observed by Skiles (2014) in daily vertical profiles of optical snow grain size over the 2013 ablations season in Senator Beck Basin.

Second, when there is an actual inversion of grain sizes with elevation, we suggest this is accounted for by rapid melt induced

by high impurity content at the surface. This was also observed in the daily field observations in 2013 by Skiles (2014), when the dust layers coalesced at the surface and skies were clear both grain sizes and density in the surface layers decreased and surface roughness increased. These observations, Painter et al. (2013) and the results in this paper suggest that the intensification of snowmelt by dust in the visible wavelengths results in destructive metamorphism near the snow-atmosphere interface and snowmelt infiltration with refreezing and snow grain coarsening at depths of ca. 2–10 cm. The observed snow grains in the

near-surface layer can therefore be smaller under intense snowmelt. The results could also suggest that sublimation could form frost flower-like crystals on the surface of a refreezing surface layer resulting in an increase of SSA and a decrease in snow grain size  (Domine et al., 2005).

In addition, our analyses confirm the inverse correlation between the snow albedo and grain size distribution  (Flanner and Zender, 2006). Smaller snow grains have smaller absorbing path lengths and smaller single-scattering asymmetry parameters

(more side-scattering), which combined lead to higher snow albedo. We find in our results a similar inverse correlation between snow albedo and RF. The quantitative relationships between radiative forcing and the spatial and temporal parameters can vary with respect to snow albedo due to the differences in the local instantaneous solar irradiance.

## 6   Conclusions

Snow cover across the worlds mountainous regions is important for both regional climate and hydrology. Snow albedo, itself

controlled by impurity content and grain size, is particularly relevant for understanding melt initiation and melt rates. Ongoing consistent measurements of snow albedo only occur at six snow energy balance sites across the western US, with even sparser or nonexistent measurements globally (Painter et al., 2012b). The analysis of the snow remote sensing products presented here, derived from a series of imaging spectroscopy data, provides new insights into the spatial distribution of snow albedo and snow





optical properties at five time steps across the accumulation/ablation transition in the Sierra Nevada and across the ablation season in the Rocky Mountains. The focus of most of the publications on snow property retrievals from imaging spectrometer data (see Sect. 3) has been on algorithm development and technology demonstration. Here we take the next step, utilizing the proven data and algorithms to assess spatial and temporal patterns. While some of our results are intuitive, such as smaller grain

sizes on north facing slopes and the inverse relationship between grain size and albedo, other were counterintuitive, including faster grain growth rates at higher elevations and higher grain sizes at high elevations relative to lower elevations late in the ablation season. This is the first time we have been to analyze patterns like these at this spatial and spectral scale.

The retrieval of impurity radiative forcing is especially relevant as efforts are now in place to reduce short-lived climate pollutants, such as black carbon, brown carbon, methane, tropospheric ozone and some hydrofluorocarbons by the Climate and

Clean Air Coalition managed by the United Nations Environmental Programme (Shindell et al., 2011). It will become critical to monitor these short-lived climate pollutants with respect to their spatial distribution on the local to global scale, as well as their change in time. Additional studies of the spatial and temporal distribution of impurity content in snow and ice of large mountain ranges, boreal and arctic regions are needed to enable us to improve our understanding of snow in the framework of complex climate-terrain feedbacks. The application of our methods and algorithms to airborne and upcoming spaceborne

imaging spectrometer data, such as the Hyperspectral Infrared Imager (HyspIRI), are potential tools to address this important question.

In addition, further application of the data and algorithms described here and in Painter et al. (2013) could provide valuable insights for water resources management to increase hydropower generation efficiency and to help better secure drinking water availability. As an example, NASA's Airborne Snow Observatory (Painter et al., 2015) applies these retrieval methods

operationally to quantify snow optical properties in the full Tuolumne River basin, Sierra Nevada, California, on a weekly basis throughout the ablation periods of years 2013 to 2015. In this case, additional LiDAR data are being used to also quantify snow hydrology parameters, such as snow depth and SWE.

*Author contributions.*  F.C. Seidel wrote and applied the retrieval codes to derive the snow products, which are based on original algorithms by T.H. Painter. K. Rittger prepared the AVIRIS and snow pillow data. S. M. Skiles contributed content related to snow hydrology. F.C.

Seidel prepared the manuscript with contributions from all co-authors.

*Acknowledgements.*  This research was performed at the Jet Propulsion Laboratory, California Institute of Technology, under a contract with the National Aeronautics and Space Administration. Thanks to Dr. Noah Molotch as the investigator for the Southern Sierra Snow Survey flights with the ER-2, to the AVIRIS and ER-2 teams, and Dr. Peter Kirchner for useful discussions. Copyright 2016 California Institute of Technology. All Rights Reserved 2016.



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



**Table 1.** Data parameters of the study areas derived as a mosaic from the subsets of flight lines (see Table 2 and Fig. 1).

| Sierra Nevada, CA. 2009 | |
| --- | --- |
| Dates | 27 Feb., 05 Mar., 04 Apr., 07 May, 08 May |
| Pixel Size [m] | 14.5 |
| Image Area [ $km^2$,pixels] | $252.03067, 1.2 \cdot 10^6$ |
| Elevation Min./Mean/Max. [m] | 1140 / 2575 / 3533 |

| Rocky Mountains, CO. 2011 | |
| --- | --- |
| Dates | 13 May, 16 May, 09 Jun., 15 Jun., 23 Jun |
| Pixel Size [m] | 13.8 |
| Image Area [ $km^2$,pixels] | $54.721982, 2.9 \cdot 10^5$ |
| Elevation Min./Mean/Max. [m] | 2950 / 3605 / 4111 |



**Table 2.** Summary of the individual AVIRIS scenes used in this paper. The scene number corresponds to the AVIRIS 'run ID'. The aircraft altitude, heading, and the terrain elevation are averages per scene. The solar geometry is valid at the center of a scene. Water vapor is retrieved using ATCOR-4 (see Sect. 3).

| AVIRIS | | Aircraft | | Mean | Pixel | Solar | | Water |
|---|---|---|---|---|---|---|---|---|
| Flight Date | Scene Nr. | Altitude [km] | Heading [deg.] | Elevation [km] | Size [m] | Zenith [deg.] | Azimuth [deg.] | Vapor [cm] |
| **Sierra Nevada, CA. 2009** | | | | | | | | |
| 27 Feb | 6 | 19.799 | 354 | 2.547 | 14.1 | 48.3 | 205.7 | 0.32 |
| | 8 | 19.836 | 185 | 2.925 | 13.9 | 56.1 | 205.4 | 0.23 |
| 05 Mar | 5 | 19.806 | 181 | 2.056 | 14.4 | 45.7 | 155.3 | 0.22 |
| | 6 | 19.765 | 000 | 2.554 | 14.0 | 44.7 | 158.7 | 0.13 |
| | 7 | 19.773 | 181 | 2.879 | 13.7 | 44.1 | 162.7 | 0.09 |
| 04 Apr | 5 | 19.799 | 182 | 2.082 | 14.4 | 41.0 | 131.6 | 0.16 |
| | 6 | 19.674 | 359 | 2.552 | 13.9 | 39.5 | 134.4 | 0.10 |
| | 7 | 19.902 | 181 | 2.911 | 13.8 | 38.5 | 137.3 | 0.07 |
| | 8 | 19.796 | 000 | 1.865 | 14.5 | 37.4 | 139.7 | 0.24 |
| 07 May | 6 | 14.262 | 348 | 3.020 | 9.0 | 23.4 | 144.0 | 0.30 |
| | 7 | 14.274 | 193 | 2.650 | 9.3 | 22.4 | 148.6 | 0.43 |
| | 8 | 14.273 | 349 | 2.277 | 9.3 | 21.5 | 154.4 | 0.62 |
| 08 May | 5 | 14.141 | 343 | 3.178 | 8.9 | 27.6 | 129.0 | 0.36 |
| | 6 | 14.154 | 194 | 2.642 | 9.2 | 26.4 | 132.2 | 0.55 |
| | 7 | 14.154 | 345 | 2.383 | 9.2 | 25.0 | 136.4 | 0.73 |
| | 8 | 14.144 | 195 | 2.035 | 9.6 | 23.9 | 140.7 | 0.89 |
| | 9 | 14.157 | 346 | 2.715 | 9.2 | 22.6 | 146.1 | 0.58 |
| **Rocky Mountains, CO. 2011** | | | | | | | | |
| 13 May | 5 | 6.496 | 352 | 3.560 | 2.1 | 19.6 | 173.0 | 0.25 |
| | 6 | 6.536 | 354 | 3.526 | 2.1 | 19.5 | 182.9 | 0.25 |
| | 7 | 6.533 | 354 | 3.683 | 2.1 | 19.9 | 192.6 | 0.21 |
| 16 May | 7 | 19.866 | 180 | 3.237 | 13.5 | 20.5 | 153.5 | 0.22 |
| 09 Jun | 12 | 20.170 | 179 | 3.260 | 13.7 | 15.0 | 179.1 | 0.33 |
| 15 Jun | 12 | 20.216 | 180 | 3.196 | 13.8 | 20.8 | 129.7 | 0.51 |
| 23 Jun | 5 | 20.139 | 001 | 3.271 | 13.7 | 38.3 | 102.0 | 0.54 |





**Figure 1.** Geographic extend of the AVIRIS image data. The study area is shown hatched. It is based on a mosaic of subsets, such that a common area exists for all flight dates. The elevation model is hill shaded in the background with an illumination corresponding to the mean solar position for all dates.



**Figure 2.** Histograms of the terrain aspect (top) and elevation (bottom) for the two study sites, Kings/Kaweah Basin Sierra Nevada, CA. (left) and Uncompahgre Basin Rocky Mountains, CO. (right). The values are normalized by the size of the study area given in Table 1. The peak occurrences of terrain aspect give an indication to the main drainage direction of a river basin.



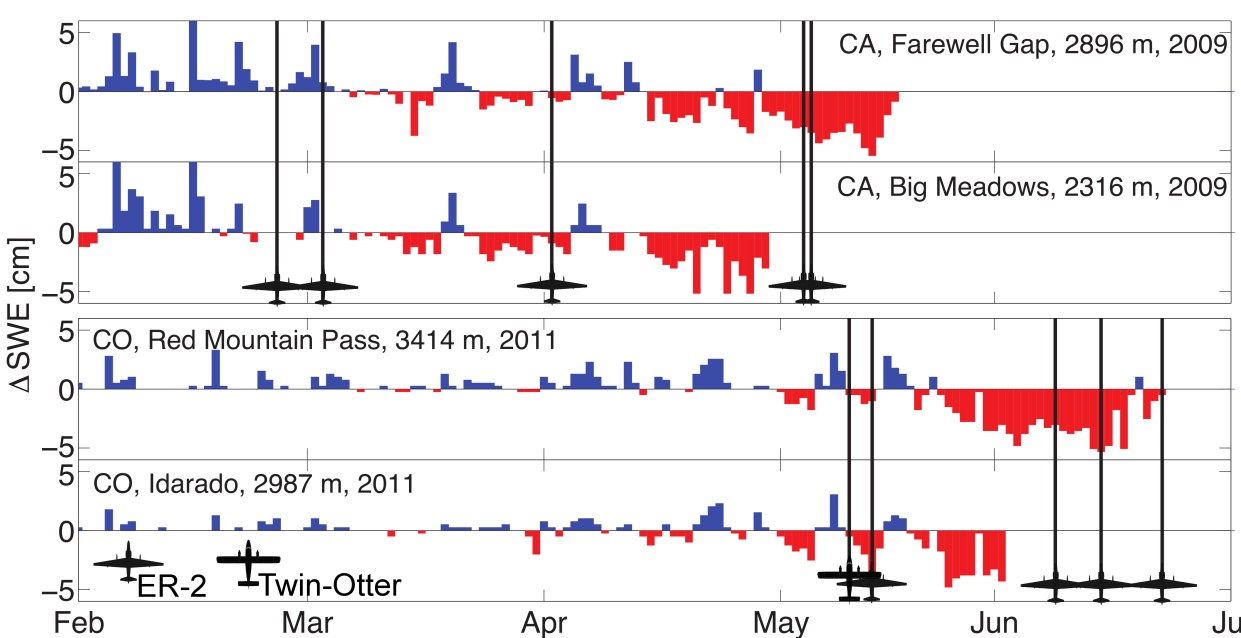

**Figure 3.** Snow accumulation (blue) and snowmelt (red) indicated with the differential Snow Water Equivalent (Δ SWE) with respect to time. The AVIRIS flight dates are marked with black vertical lines and a symbol indicating the ER-2 or Twin-Otter airborne platform. The upper plot shows snow sensor pillow data from two stations representing a higher and a lower location in the Sierra Nevada, CA, study area in 2009. The lower plot shows the same information, but from stations close to the Rocky Mountains, CO, study area in 2011.



**Figure 4.** Maps of snow grain size retrieved using Eq. (4) for three dates during the melt period in the Sierra Nevada (upper panel), and the Rocky Mountains (lower panel). The upper and lower map represent 252 km$^2$ and 55 km$^2$, respectively (Table. 1).







**Figure 5.** Same as Fig. 4 but for snow albedo.





**Figure 6.** Same as Fig. 4 but for snow radiative forcing.



**Figure 7.** Time-series of snow cover, albedo, grain size and radiative forcing in the Sierra Nevada, CA., USA in 2009 (upper panel), and in the Rocky Mountains, Co., USA in 2011 (lower panel). The snow cover is normalized by the size of the study area and the other values represent the average value per date surrounded by the corresponding standard deviation.



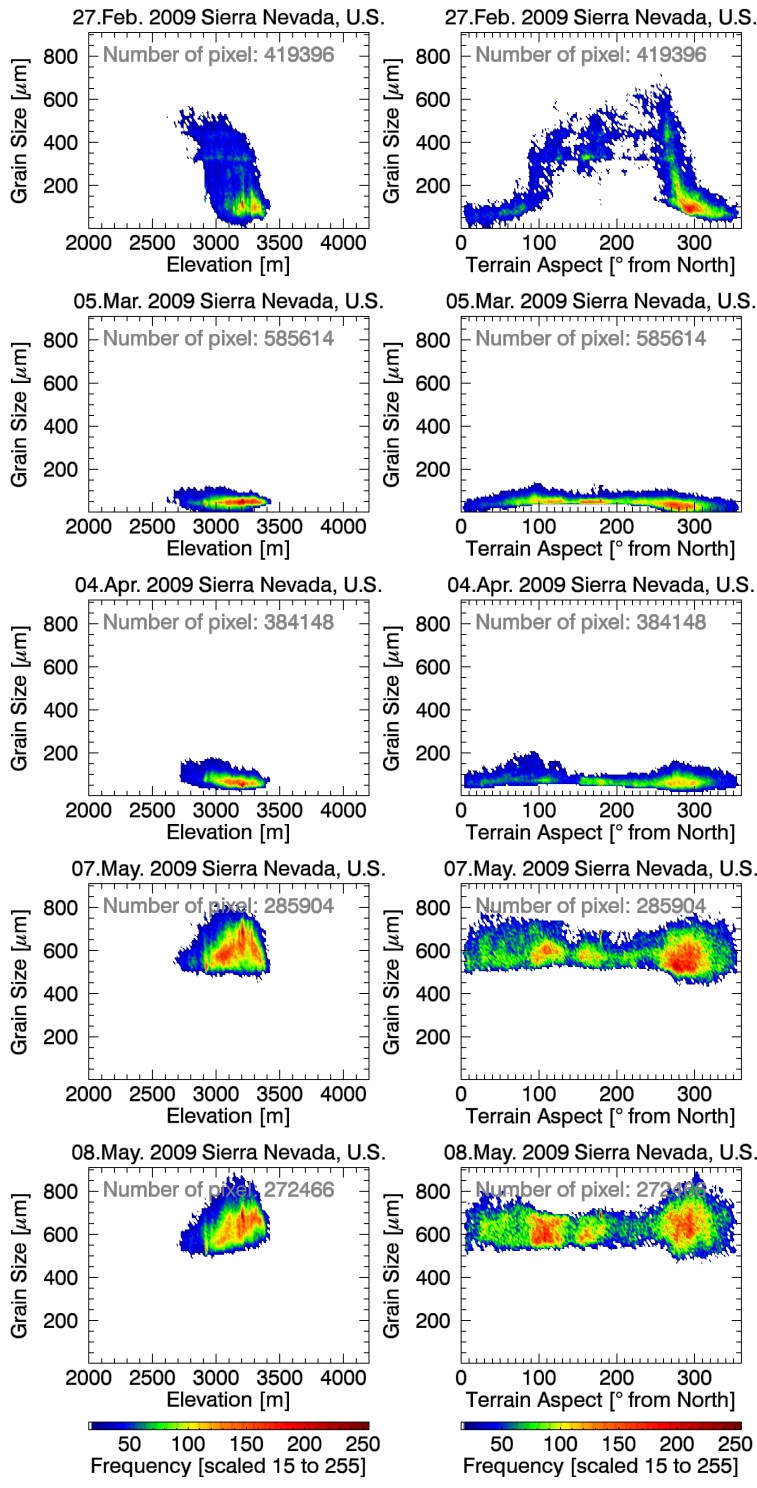

**Figure 8.** Snow grain size with respect to terrain elevation (left) and aspect (right) in the Sierra Nevada in spring 2009.



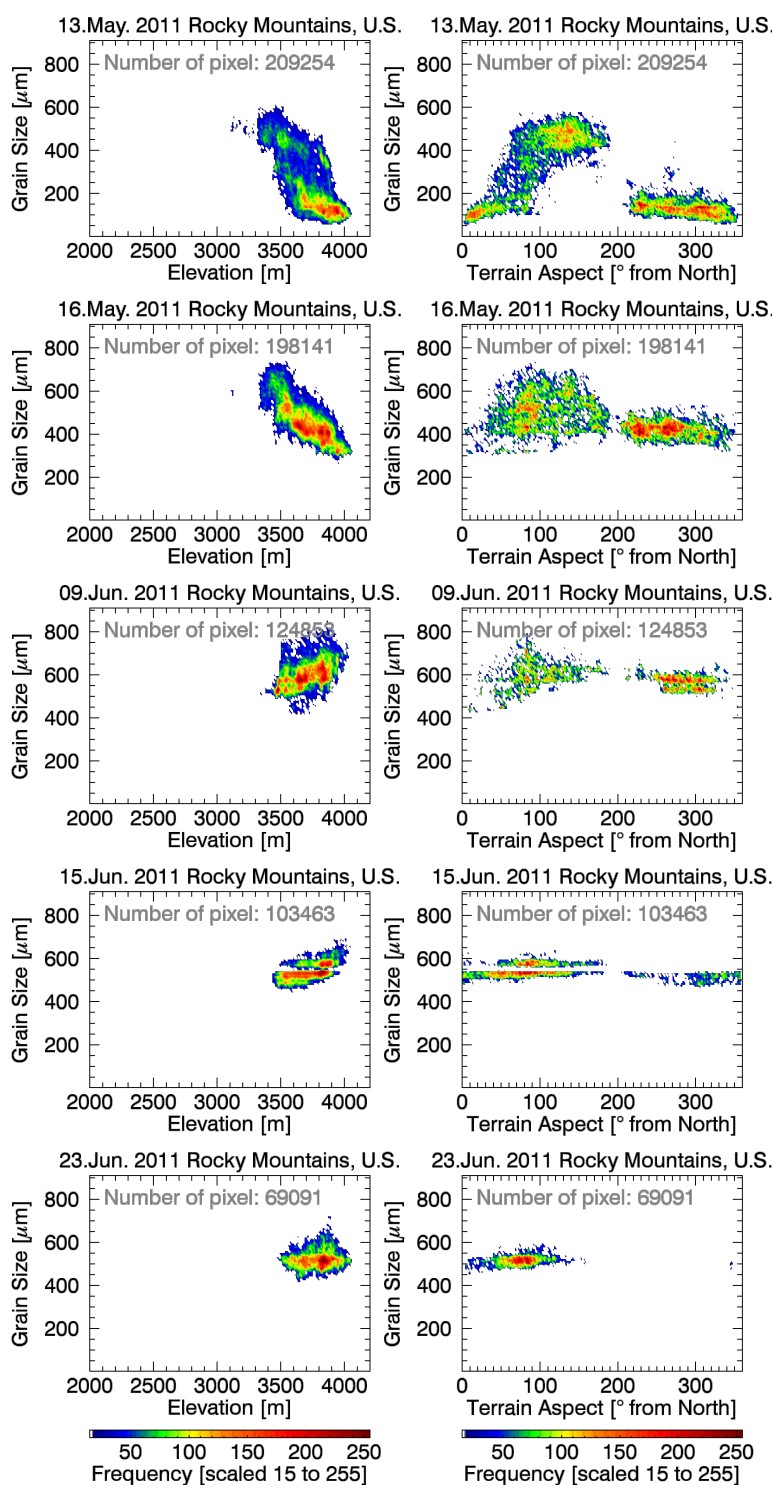

**Figure 9.** Same as Fig. 8 but for the Rocky Mountains in spring 2011.





**Figure 10.** Snow albedo along a North–South (left), and West–East transect (right) through the middle of the observed Sierra Nevada area. The values are smoothed with a boxcar average of 30 pixels. The snow-free surface is shown in black.





**Figure 11.** Same as Fig. 10 but for the Rocky Mountains.





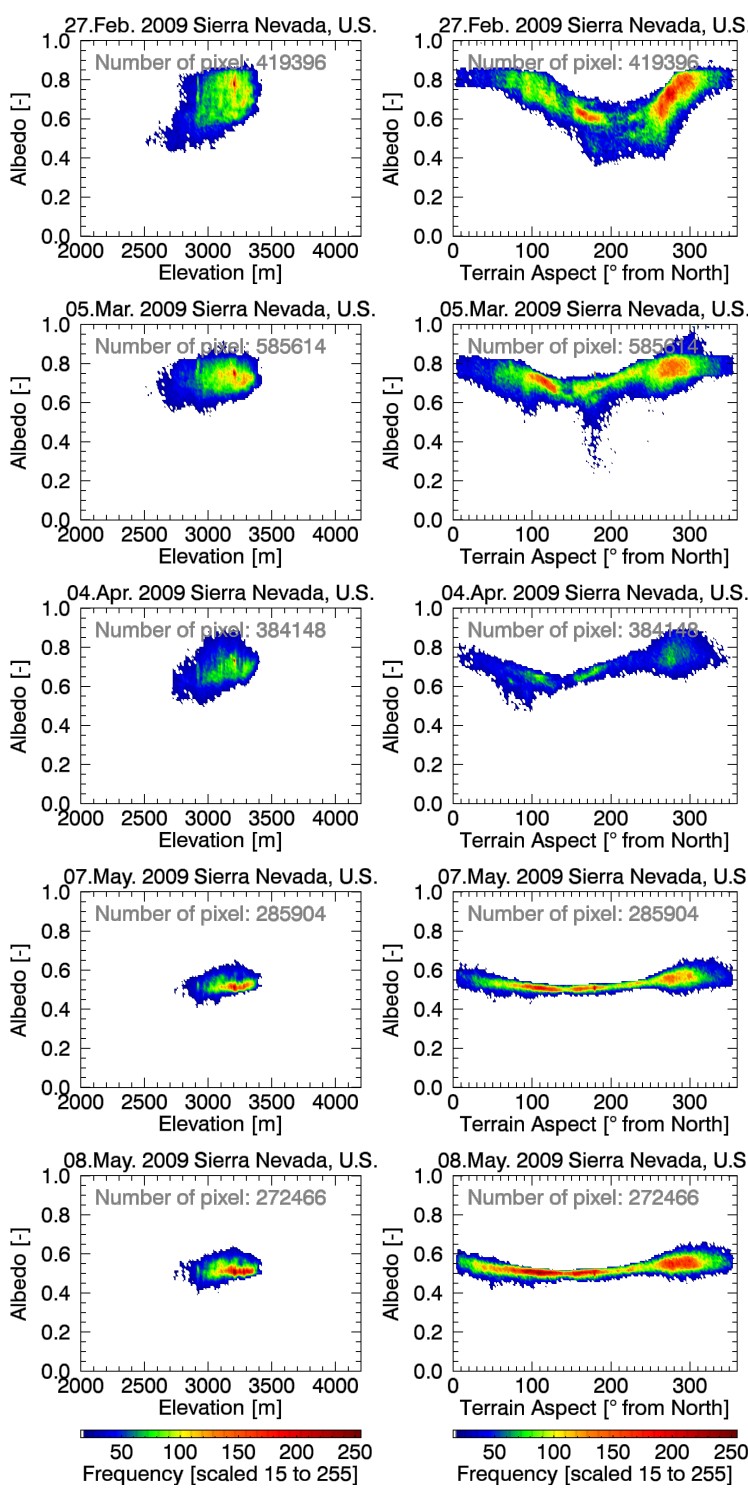

**Figure 12.** Snow albedo with respect to terrain elevation (left) and terrain aspect (right) in the Sierra Nevada in spring 2009.



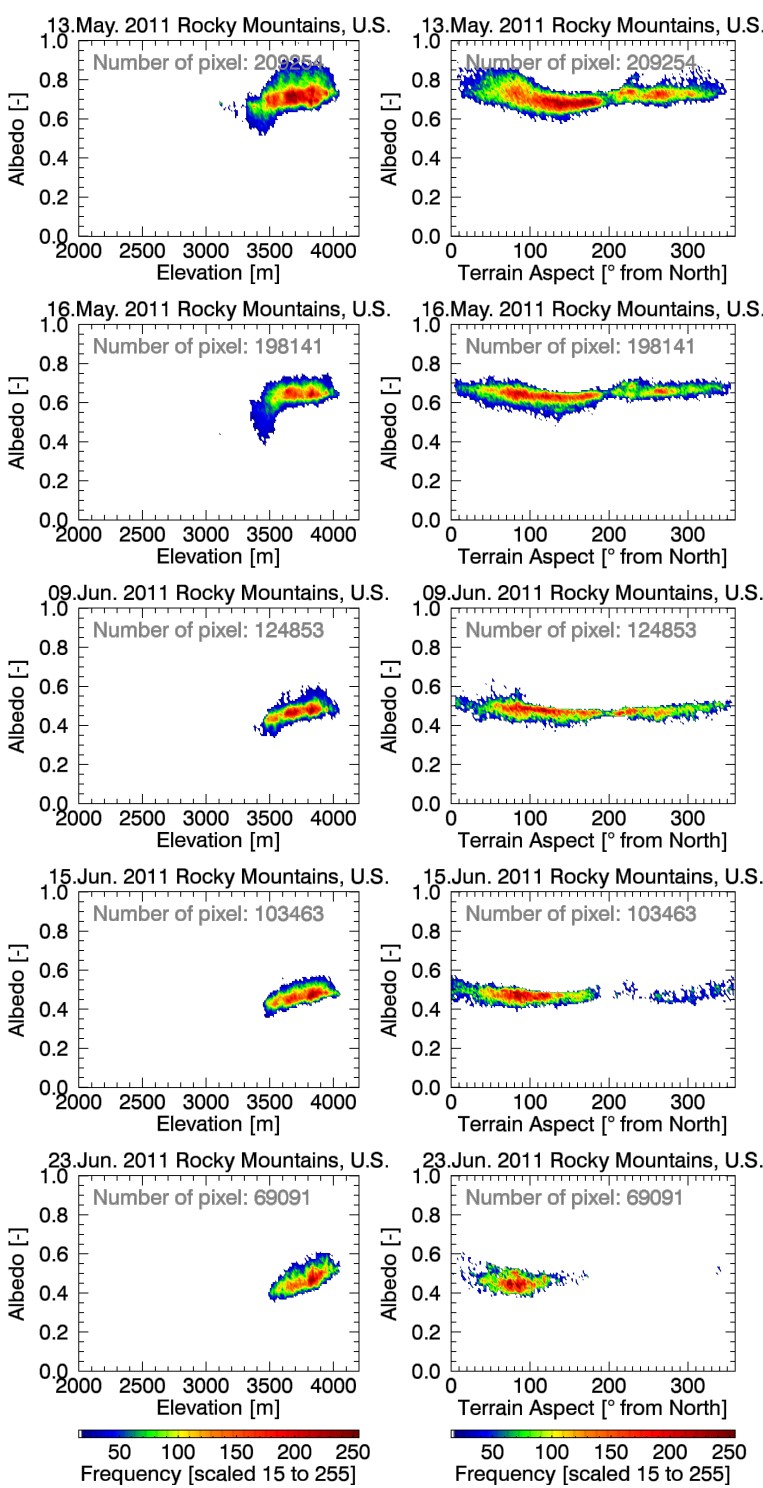

**Figure 13.** Same as Fig. 12 but for the Rocky Mountains in spring 2011.



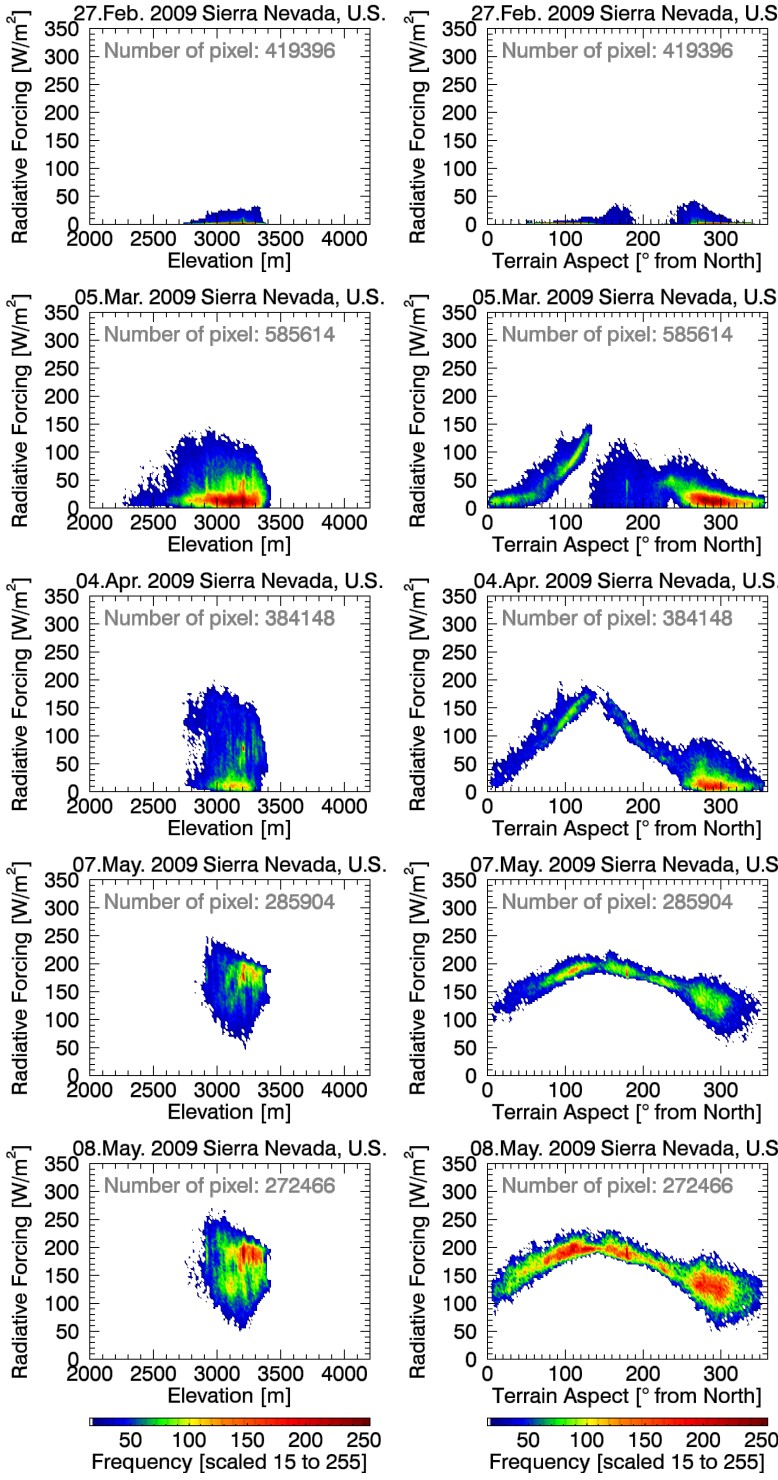

**Figure 14.** Radiative forcing of snow impurities with respect to terrain elevation (left) and aspect (right) in the Sierra Nevada in spring 2009.







**Figure 15.** Same as Fig. 14 but for the Rocky Mountains in spring 2011.