# Peer review of "Case study of spatial and temporal variability of snow cover, grain size, albedo and radiative forcing in the Sierra Nevada and Rocky Mountain snowpack derived from imaging spectroscopy"

_The Cryosphere, 2015_

## Referee Comment (RC1) · Anonymous Referee #1 · 16 Feb 2016

This study represents a nice demonstration of the utility of air-borne hyperspectral measurements for inferring spatial distributions of important snow properties. As the authors note in their conclusions, previous studies utilizing AVIRIS data have mostly focused on algorithm development and demonstrations of feasibility, whereas this study goes a bit deeper to describe actual variability in snow properties determined from a series of flights over the Rockies and Sierra Nevada. Most of the analysis shows variability in snow properties with elevation, aspect, and seasonal progression that are to be expected from basic physical arguments. Other measurements, however, show less intuitive distributions, such as the *positive* correlation between snow grain size

and elevation seen in the Rockies during June, as well as the decrease in spatial variability of snow albedo observed during progression of the melt season. Such findings are really only feasible with remote sensing measurements, thus demonstrating their value. The manuscript is well-written and logically organized. Below are a few general comments and several minor comments that should be addressed prior to publication in The Cryosphere. Overall, however, I find the study and manuscript to be in good shape.

Major comments:

Data presented in Figure 9, panel h (15 June, terrain aspect plot) appear to show no grain size measurements between roughly 500 and 550$\mu$m, with many measurements bracketing either side of this range. Is this gap an artifact of the retrieval algorithm (e.g., coarse discretization)? The gap should be explained, in any case, as it does not appear to be natural.

Second, how exactly were surface slope and aspect determined? Were these determined exclusively from the (snow-free) DEMs, or were they determined directly from the AVIRIS measurements? The normal to the surface can obviously change with snow accumulation, as valleys are filled, snow drifting occurs, etc. If DEMs were used to determine slope and aspect, please elaborate on the magnitudes of error that could result from variable (or inaccurate) slope and aspect. If they were determined from the AVIRIS measurements, please explain how.

The term "radiative forcing" is used frequently, and it eventually becomes clear that this refers to surface radiative forcing from light absorbing impurities, but in general "radiative forcing" is ambiguous. It could also apply, e.g., to grain size effects. At first reference (in the abstract), and perhaps throughout the paper, this should instead be referred to with something like 'impurity radiative forcing' or 'LAISI radiative forcing'. It should also be mentioned early that the reference plane for the forcing is the surface (rather than top-of-atmosphere or tropopause or aircraft altitude).

Related to the point above, do the radiative forcings represent local noon values or daily averaged values (or something different)? Please clarify.

Minor comments:

p1,13: Please specify the time period over which albedo decreases from 0.7 to 0.5. Presumably this is during the ablation period, but the ablation timeframe for each location should also be mentioned in the abstract.

p2,13: More precisely, this is the radius of a sphere that has specific surface area of SSA. For a polydisperse size distribution of spheres with collective specific surface area of SSA, it is the effective radius of that distribution.

p3,25: Wording (run-on sentence).

p4,3: particular -> particularly

p4,6: "end year dust concentrations" - Are these concentrations at the top of the snow, or are they column-averaged concentrations? If the former, please specify the thickness of snow over which the concentrations apply.

p4,13-16: Sentence structure problems.

p5,22: Why are '$\approx$' symbols used? Which wavelengths were actually used to determine the NDSI? And, were single AVIRIS bands used, or were spectral averages taken over multiple bands? Please clarify.

p6,6-8 (equation 4): If the ice absorption feature is centered at $1.27\mu$m, why is the upper bound of the integral at precisely $1.27\mu$m? (i.e., why not integrate over the feature, rather than up to it?). Presumably this is explained in one of the cited publications, but a very brief explanation for this may be warranted here.

Equation 5: I would speculate that this equation occasionally produces an observed albedo larger than 1, especially outside of the $1.17$-$1.27\mu$m grain size calibration window. Does this ever happen, and is it necessary to cap the albedo at 1?

Section 3.6 discussion: This is fine (and helpful), but it should be clarified (e.g., in section 1) that the optical grain radius used here is a sphere-equivalent optical radius.

p7,11: Again, is the radiative forcing uncertainty defined with respect to daily-mean forcing or local noon instantaneous forcing?

p8,3-7: Please rework this passage for grammar and clarity.

p9,11: "In general the albedo increase with elevation is more distinct at the Sierra Nevada study area than in the Rocky Mountain study area." - This is interesting. Do you have any hypotheses for why?

p9,24-25: Again, are these instantaneous RF values?

p10,1: "all of the dust" -> maybe "nearly all of the dust"?

p10,15: Are you arguing here that earlier melt out on the south-west face reduces the time over which forcing can operate, therefore leading to greater forcing on the south-east face? Please clarify the relationship being described between processes on the south-east and south-west slopes.

p11,16: Please check the wording in this section.

p11,19-20: "The observed snow grains in the near-surface layer can therefore be smaller under intense snowmelt." - This is quite interesting!

Figure 6 caption: Following previous comments, please clarify the source and temporal averaging domain of this forcing.
* * *

---

## Referee Comment (RC2) · M. Dumont (Referee) · 22 Feb 2016

This paper presents a really interesting data set of snow properties (optical grain size, broadband albedo and radiative forcing due to impurities) retrieved from hyperspectral airborne data over two mountainous areas. The areas and the retrieval methods are first described. The authors then present the results in term of spatial and temporal variations of the above mentioned snow properties. Note that the retrieval method has been described and evaluated in a previous paper (Painter et al., 2013).

This study is well written and fits well with The Cryosphere scope. The data set provided by the authors is quite unique. The study demonstrates the benefits of using high resolution airborne optical data to study in detail temporal and spatial variations of snow properties. It offers a new insight for the study of snow properties variability avoiding the limitations of point measurements. I think however that several questions should be addressed before it can be published.

**Main Comments**

1/ My first main comment focused on the grain size retrieval accuracy in presence of liquid water. It is stated section 3.3 that the range of wavelengths used in the grain size retrieval is 1.17 to 1.27 $\mu$m allowing to reduce the "bias (induced) by liquid water" with reference to Painter et al., 2013. In Painter et al., 2013, it is on the contrary stated that the retrieval is done using 1.03 to 1.06 $\mu$m wavelengths (section 4.3) and that the "technique could be less sensitive to biases due to liquid water and water vapor". It is also stated that this assumption is not based on field measurements. Are the range of wavelengths used similar in this study and in Painter et al., 2013 or not ? In my mind, the assumption concerning the limited impact of liquid water on the retrieval should be discussed and demonstrated in more depth.

For example, considering figure 1 in Gallet et al., 2014a or Green et al., 2002, there are some differences between the ice and water refractive indices in the considered range of wavelengths. This can induce some errors on the grain size retrieval and should probably be included in the discussion. The algorithms developed in Green et al., 2002 can also probably be used to investigate in more detail the effect of liquid water of the grain size retrieval and also to identify on the AVIRIS data snow with liquid water. I think this point is really crucial to infer the uncertainty of grain size retrieval and to strengthen the discussion on the grain size spatial and temporal variations presented in this study.

*Gallet, J.-C., Domine, F., and Dumont, M.: Measuring the specific surface area of wet snow using 1310 nm reflectance, The Cryosphere, 8, 1139-1148, doi:10.5194/tc-8-1139-2014, 2014a.*

*Green, R. O., Dozier, J., Roberts, D., Painter, T. (2002). Spectral snow-reflectance models for grain-size and liquid-water fraction in melting snow for the solar-reflected spectrum. Annals of Glaciology, 34(1), 71-73.*

2/ The second main comment focuses on the grain size/elevation inverse relationship in case of large impurity load discussed in section 5. I think this is a really interesting finding and it would result in a negative feedback. It also has to be linked to other recent studies showing that light-absorbing impurity can reduce the density of melting snow (Meinander et al., 2014) or studying the effect of soot on anisotropy factor (Peltoniemi et al., 2015) However, the discussion should be strengthen to be more convincing. The first thing is that the mean decrease of grain size is roughly 50 $\mu$m (figures 7 and 9) which is close to the accuracy of the grain size retrieval method (the one given in Painter et al., 2013). Consequently, I think before discussing the possible physical causes of the grain size/elevation inverse relationship, the authors should probably discuss the significance of the grain size decrease relatively to differences sources of errors. There are several sources of errors that might have to be taken into consideration e.g. : a/ grain size retrieval accuracy in presence of liquid water (see my previous comment) b/ the fact that as stated by the authors the surface roughness is high. Consequently the HDRF derived from spherical model are probably more anisotropic that the natural surface inducing a larger error on the grain size for the rough surface than for flat one. c/ Peltoniemi et al., 2015 (see full reference below) also demonstrated that under large impurity load, the impurity sank in the snow and largely modify the HDRF . . . this also leads to an error in the grain size retrieval. d/ Finally, as discussed in Skiles PhD thesis (e.g. chapter 3, figure 1), it is possible that for really high impurity content, the snow reflectance is more dust-like than snow-like in the IR (and then higher in the NIR not due to snow grain size but to impurity particles). This effect seems to be overestimated by SNICAR (compared to field measurements) but it can still affect the grain size retrieval and results artificially smaller grain size.

Lastly, the processes leading to a decrease of snow grain size could probably further
discussed : the presence of a large content of impurity largely modifies (and shortens) the light penetration depth so that the energy is absorbed closer to the surface. The temperature profile and the heat and vapour transfer are consequently modified leading to modified metamorphism. Note that the formation of sublimation crystals has also been discussed in Antarctica (Gallet et al., 2014b, see full reference below).

*Meinander, O., Kontu, A., Virkkula, A., Arola, A., Backman, L., Dagsson-Waldhauserová, P., Järvinen, O., Manninen, T., Svensson, J., de Leeuw, G., and Leppäranta, M.: Brief communication: Light-absorbing impurities can reduce the density of melting snow, The Cryosphere, 8, 991-995, doi:10.5194/tc-8-991-2014, 2014.*

*Peltoniemi, J. I., Gritsevich, M., Hakala, T., Dagsson-Waldhauserová, P., Arnalds, Ó., Anttila, K., Hannula, H.-R., Kivekäs, N., Lihavainen, H., Meinander, O., Svensson, J., Virkkula, A., and de Leeuw, G.: Soot on Snow experiment: bidirectional reflectance factor measurements of contaminated snow, The Cryosphere, 9, 2323-2337, doi:10.5194/tc-9-2323-2015, 2015.*

*Gallet, J.-C., Domine, F., Savarino, J., Dumont, M., and Brun, E.: The growth of sublimation crystals and surface hoar on the Antarctic plateau, The Cryosphere, 8, 1205-1215, doi:10.5194/tc-8-1205-2014, 2014b.*

3/ The third main comment focuses on the way the data are presented in Figures 8 and 9 with respect and elevation. I think the figures really nicely described the variations of the snow parameters with respect to topographic parameters. However, especially at the end of the season, when the snowpack remains mainly on shaded slopes this way to present the data could induce a bias in the interpretation. In other words, I mean that it is, for example, not straightforward from Figure 9, 3rd row that the inverse grain/size elevation relation ship is not a bias due to uneven distribution of snow cover with elevation and aspect. Consequently, my suggestion would be to plot the value of mean grain size (colour) as a function of aspect (x-axis) and elevation (y-axis) with a representation of the RMSE or the number of points in each topographic class (transparency or size

of the dots).

4/ The last main comment focuses on the uncertainties analysis provided in the paper. Though I perfectly agree that uncertainty analysis is not the main aim of the paper, I think that a deeper analysis of the errors could help to strengthen the conclusions of the paper as I explained in the previous comments. The authors refers to the uncertainties of the retrieval method provided in Painter et al., 2013 as being 25um for grain size, 0.0001 (there must be a typo there) for snow albedo and 4 Wm2 for radiative forcing. In Painter et al. 2013, the uncertainties given for albedo is 0.001-0.004 (based on two measurements only. . .), 2.1 +-5.1 Wm2 for radiative forcing, and the uncertainty of the grain size retrieval is an assumption. In Painter et al., 2013, only one date was available for AVIRIS data but with all the dates used in the study, I am wondering if the uncertainty data could used more broadband albedo measurements to strengthen the uncertainty analysis. Section 4.5. presents really interesting results regarding the comparison of two images acquired only one day apart from each other but it would worth a more detailed analysis. Are the given values mean values ? What are the spreads of the difference ? In my mind, the uncertainty analysis should be more careful, describing which values are assumed, what is the significance of the different values (how many measurements are used to derive the uncertainty). . .

**Specific Comments**

Page 2, line 13 : Maybe the authors can be a bit more explicit why the SSA /optical grain radius relationship is not a strict equality.

Section 3.3 . The modeled HDRF must account for the diffuse to total incoming irradiance ratio. Therefore it could worth explaining how this ratio is calculated. For this section, please see also my main comment 1/.

Page 7, Equation 7 : If c is meant to account for imperfect terrain corrections, c probably also have a second order dependency to wavelength. This could be maybe detailed in the paper.

Page 7, line 13 : *This assumption becomes more valid with grain growth associated with evolving metamorphism and rounding of the snow crystals.* Rounded forms do not necessarily means perfect spheres, and spheroids optical properties might significantly differ from spheres optical properties (e.g. Libois et al., 2013, full ref below, fig1). The differences between spheres and spheroids optical properties might even be larger that the ones between spheres and other snow crystals optical properties. Consequently, in my mind, this sentence should be either removed or further justified.

*Libois, Q., Picard, G., France, J. L., Arnaud, L., Dumont, M., Carmagnola, C. M., and King, M. D.: Influence of grain shape on light penetration in snow, The Cryosphere, 7, 1803-1818, doi:10.5194/tc-7-1803-2013, 2013.*

Page 8, lines 9-13 : The mean broadband albedo varies not only with the snow properties but also with the illumination geometry and consequently the date and time of acquisition. This should probably be discussed while presenting the results of Figure 7.

Section 4.3. Same comment as above. Snow albedo depends not only on the snow properties but also on the illumination conditions. This has, in my mind, to be taken into account in the discussion regarding the spatial variations of albedo.

**Minor Comments**

Page 2, line 16 : Changes is surface roughness also cause changes in snow albedo

Page 4, line 22 : "retrieved columnar water vapor", maybe the authors could explain how, and from which data and add a reference.

Page 8, line 18 : Surface snow grain size can also be affected by wind effects.
* * *

---

## Author Response (AR1)

**TC-2015-196**

**Combined Author Comments**

We are very grateful for the thoughtful comments by both reviewers. They helped considerably to improve our manuscript.

Below, please find the reviewer comments in serif font and our **replies in bold sanserif font**.

We use a ***bold sanserif italic font*** to identify new or changed text.

We applied few minor improfements to enhance the readability of the manuscript.

Please find all changes highlighted ( and new) in the appended manuscript.

**Reply to Comments by Anonymous Referee #1**

This study represents a nice demonstration of the utility of air-borne hyperspectral measurements for inferring spatial distributions of important snow properties. As the authors note in their conclusions, previous studies utilizing AVIRIS data have mostly focused on algorithm development and demonstrations of feasibility, whereas this study goes a bit deeper to describe actual variability in snow properties determined from a series of flights over the Rockies and Sierra Nevada. Most of the analysis shows variability in snow properties with elevation, aspect, and seasonal progression that are to be expected from basic physical arguments. Other measurements, however, show less intuitive distributions, such as the *positive* correlation between snow grain size and elevation seen in the Rockies during June, as well as the decrease in spatial variability of snow albedo observed during progression of the melt season. Such findings are really only feasible with remote sensing measurements, thus demonstrating their value. The manuscript is well-written and logically organized. Below are a few general comments and several minor comments that should be addressed prior to publication in The Cryosphere. Overall, however, I find the study and manuscript to be in good shape.

Major comments:

Data presented in Figure 9, panel h (15 June, terrain aspect plot) appear to show no grain size measurements between roughly 500 and 550μm, with many measurements bracketing either side of this range. Is this gap an artifact of the retrieval algorithm (e.g., coarse discretization)? The gap should be explained, in any case, as it does not appear to be natural.

**This gap is indeed an artifact. It is caused by non-unique solutions of Equation 4 where retrievals between 540 and 570 *μ*m are lumped together, leaving a gap in between those values. It appears to be a rare case where the modeled HDRF at this given solar geometry and terrain can fit the data for multiple snow grain sizes. This effect is accentuated by the plotting routine, which removes 'outliers' where less than 15 pixels were found with the same combination of X and Y values.**
**More work would be required to fully understand and overcome this artifact by improved inversion methods. However, given all of our snow grain size retrievals, we think that this a rare and isolated case. Further, we do not think this artifact affects any of our conclusions in this paper. We thus decided to add a short description to the caption of Figure 9:** *"Note: A rare retrieval artifact is found on 15 June 2011, where Eq. 4 has no unique solution within a snow grain size range of 540 to 570 μm causing a gap in the corresponding plots."*
* * *
Second, how exactly were surface slope and aspect determined? Were these deter- mined exclusively from the (snow-free) DEMs, or were they determined directly from the AVIRIS measurements? The normal to the surface can obviously change with snow accumulation, as valleys are filled, snow drifting occurs, etc. If DEMs were used to determine slope and aspect, please elaborate on the magnitudes of error that could result from variable (or inaccurate) slope

and aspect. If they were determined from the AVIRIS measurements, please explain how.

**As mentioned in Sect. 2.1 (page 3, lines 2-10), we use the DEM, slope, and aspect data from the AVIRIS data products (specifically from the *_ort_igm and the *_obs_ort files), which are based on the USGS NED 1 arc second (~30 m) and converted from the raw GPS ellipsoidal elevation to orthometric elevations via the use of the NGA EGM96 global geoid model. From this, we determine the slope and aspects using standard routines provided by the ENVI software package, which are based on the following reference: "Wood, Joseph The Geomorphological Characterization of Digital Elevation Models, Ph. D. Thesis, University of Leicester, Department of Geography, Leicester, UK, 1996."**

**It is worth mentioning that the DEM in our previous study (Painter et al, 2013) was based on the very coarse GTOPO30, which indeed was source of errors. The quality and the spatial resolution of the DEM used in this study is much better. This greatly improves the determination of the pixels being in the shadow of the terrain. In fact, terrain shadows can make up to 10% of the snow covered area early in the observed season. ATCOR-4 (Richter and Schläpfer, 2014) is capable of deriving the HDRF within shadows (by neglecting the direct irradiance component), but we suspect that strong adjacency effects in snow covered mountain terrain induce large uncertainties. We therefore remove those pixels from our analysis as explained in Sect. 4.5 (page 10, line 25). In addition, we typically find a few pixels (less than about 0.1% of the snow pixels) very close to a crest affected by the DEM where image data are attributed to the wrong side of the crest. The affected retrievals are found to be clear outliers. We don't show these retrievals in the density plots due to our threshold requirement of 15 pixels retrieved within the same bin of the histogram (see also our reply above). Lastly, we don't have a snow-on surface model available for this study, as opposed to our work performed within NASA's Airborne Snow Observatory (ASO). We therefore have no choice but to ignore this effect. However, we expect this to be a small source of errors for a majority of pixels. It would be worthwhile to test this hypothesis using data from ASO for example.**

**We improved our manuscript in Sect. 2.1 by mentioning AVIRIS and in Sect. 4.5 by specifically mention the issues related to crests and to the use of a snow-off DEM. The second paragraph of Sect. 4.5 is now worded as follows:** *"Additional error sources are related to terrain shadows and the digital elevation model (Sect. 4.5). Up to 10% of the snow pixels in the February data are in shadow due to the large solar zenith angle. Snow properties can be retrieved in shadows using the diffuse irradiance. However, we suspect that large uncertainties are introduced due to adjacency effects, which are difficult to model in snow covered mountains. We therefore remove retrievals in shadow areas from our analysis, except for the snow cover calculation. Pixels very close to a crest can be associated with the wrong mountainside, which causes retrieval outliers. We found only very few snow pixels affected by this artifact and they do not show up in the histogram*

*plots (Figs. 8, 9, and 12-15) due to a threshold of 15 occurrences. Changes in slope and aspect of the surface due to snow accumulation are not taken into account in this study and further analysis would be required to quantify associated errors."*
* * *
The term "radiative forcing" is used frequently, and it eventually becomes clear that this refers to surface radiative forcing from light absorbing impurities, but in general "radiative forcing" is ambiguous. It could also apply, e.g., to grain size effects. At first reference (in the abstract), and perhaps throughout the paper, this should instead be referred to with something like 'impurity radiative forcing' or 'LAISI radiative forcing'. It should also be mentioned early that the reference plane for the forcing is the surface (rather than top-of-atmosphere or tropopause or aircraft altitude).

**Our Abstract already introduces the term LAISI on page 1, line 4 and is using it on line 12 as follows: "radiative forcing by LAISI." We think that the current Abstract sufficiently addresses this comment. In the Introduction, we generalized the term 'radiative forcing' to 'radiative effects' on page 2, line 10-11. LAISI is introduced again on page 2, line 16 and subsequently used on page 2, line 24 to clarify the term 'radiative forcing'. We added a note of clarification at this point in the manuscript that we refer to the forcing at the snowpack's surface.**
* * *
Related to the point above, do the radiative forcings represent local noon values or daily averaged values (or something different)? Please clarify.

**Our radiative forcing quantities are derived with the local instantaneous solar irradiance, i.e. the actual W/m2 entering the snowpack's surface. We added language addressing this on page 7, line 1 such that the full sentence starting on page 6, line 28 reads: "*This reduction of spectral snow albedo by LAISI is multiplied by the direct and diffuse instantaneous spectral solar irradiance entering the snow surface and integrated over the given spectrum, such that the the additional radiative flux into the snow pack is given by:*"**
* * *
Minor comments:

p1,13: Please specify the time period over which albedo decreases from 0.7 to 0.5. Presumably this is during the ablation period, but the ablation timeframe for each location should also be mentioned in the abstract.

**We now mention the location specific ablation period by changing page 1, line 8-9 to: "*Our results show a linearly decreasing snow cover during the ablation period in May and June in the Rocky Mountains and a snowfall driven change in snow*"**

*cover in the Sierra Nevada between February and May."*
* * *
p2,13: More precisely, this is the radius of a sphere that has specific surface area of SSA. For a polydisperse size distribution of spheres with collective specific surface area of SSA, it is the effective radius of that distribution.

**We added: "*or the effective radius of a grain size distribution corresponding to a specific surface area (SSA)."***
* * *
p3,25: Wording (run-on sentence).

**A period was added.**
* * *
p4,3: particular -> particularly

**Corrected.**
* * *
p4,6: "end year dust concentrations" - Are these concentrations at the top of the snow, or are they column-averaged concentrations? If the former, please specify the thickness of snow over which the concentrations apply.

**We added "*in the top 30 cm of the snow column*" for clarification.**
* * *
p4,13-16: Sentence structure problems.

**We added a period and moved the reference to increase the readability.**
* * *
p5,22: Why are '≈' symbols used? Which wavelengths were actually used to determine the NDSI? And, were single AVIRIS bands used, or were spectral averages taken over multiple bands? Please clarify.

**We now report the actual central wavelength used.**
* * *
p6,6-8 (equation 4): If the ice absorption feature is centered at 1.27µm, why is the up- per bound of the integral at precisely 1.27µm? (i.e., why not integrate over the feature, rather than up to it?). Presumably this is explained in one of the cited publications, but a very brief explanation for this may be warranted here.

**We indeed use only the first half (1.17-1.27 microns) of the feature to reduce the crosstalk from the liquid water absorption feature, which starts to dominate the signal at longer wavelengths in the presence of meltwater. We reworked the text in Sect. 3.3 to highlight this fact.**
* * *
Equation 5: I would speculate that this equation occasionally produces an observed albedo larger than 1, especially outside of the 1.17-1.27μm grain size calibration window. Does this ever happen, and is it necessary to cap the albedo at 1?

**Spectral HDRF$^{obs}$ and HDRF$^{mld}$ values can certainly be larger than 1. However, the anisotropy factor (Albedo$^{mdl}$/HDRF$^{mdl}$) scales the directional values to the hemispherically integrated albedo values, which are smaller then 1 at all wavelengths. We only derive spectral Albedo values larger than 1 where HDRF$^{obs}$ was significantly overestimated. This would happen often the snow pixels in shadows would be included in the retrieval (using the diffuse irradiance to calculate the HDRF$^{obs}$ is not sufficient due the adjacency effects, which make the observed snow brighter than expected by the model). Since we remove the shadow area, we might get some too low albedo value where the shadow mask fails to flag a shadow. Nevertheless, we retrieve very few albedo values larger than 1 close to the ridge of a steep mountainside as explained above. If the actual image pixel was in foreward scattering, but we assume backward scattering in our model, we do not compensate correctly with the anisotropy factor for the large HRDF$^{obs}$, which lead to an overestimation in the spectral Alebdo$^{obs}$. However, these few false retrievals do not show up in our plots because they either fall beyond the max value of the color table or they are ignored in the 2D histograms thanks to application of a frequency threshold of 15.**
* * *
Section 3.6 discussion: This is fine (and helpful), but it should be clarified (e.g., in section 1) that the optical grain radius used here is a sphere-equivalent optical radius.

**We already have such a statement on page 2, line 8. We modified it slightly to include the fact that we assume spheres according to your suggestion.**
* * *
p7,11: Again, is the radiative forcing uncertainty defined with respect to daily-mean forcing or local noon instantaneous forcing?

**We addressed this point as stated above.**
* * *
p8,3-7: Please rework this passage for grammar and clarity.

**We reworded this paragraph substantially.**
* * *
p9,11: "In general the albedo increase with elevation is more distinct at the Sierra Nevada study area than in the Rocky Mountain study area." - This is interesting. Do you have any hypotheses for why?

**This is actually a mistake. We wanted to state this in reverse order. We conclude this by looking at an imaginary fitting line to the data in Figs 12 and 13. We think that this might be related to the terrain itself. We fixed this in the manuscript.**

p9,24-25: Again, are these instantaneous RF values?

**We addressed this point as stated above.**

p10,1: "all of the dust" -> maybe "nearly all of the dust"?

**We changed this.**

p10,15: Are you arguing here that earlier melt out on the south-west face reduces the time over which forcing can operate, therefore leading to greater forcing on the south-east face? Please clarify the relationship being described between processes on the south-east and south-west slopes.

**We simply mean that there is no snow left on south-west facing slopes. The few remaining patches on those slopes indicate much lower RF values. However, would there still be snow on this slopes, the RF would likely be equally large as on the south-east slopes. We add the following sentence for clarification: *"The relatively low radiative forcing values on south-west to north facing slopes are likely from remnant patches of relatively clean snow."***
* * *
p11,16: Please check the wording in this section.

**We reworded this paragraph.**
* * *
p11,19-20: "The observed snow grains in the near-surface layer can therefore be smaller under intense snowmelt." - This is quite interesting!

**Yes, indeed!**
* * *
Figure 6 caption: Following previous comments, please clarify the source and temporal averaging domain of this forcing.

**We now say:** *"Same as Fig. 4 but for effective snow radiative forcing at the time of overflight."*

**Reply to Comments by M. Dumont (Referee)**

This paper presents a really interesting data set of snow properties (optical grain size, broadband albedo and radiative forcing due to impurities) retrieved from hyperspectral airborne data over two mountainous areas. The areas and the retrieval methods are first described. The authors then present the results in term of spatial and temporal variations of the above mentioned snow properties. Note that the retrieval method has been described and evaluated in a previous paper (Painter et al., 2013).

This study is well written and fits well with The Cryosphere scope. The data set provided by the authors is quite unique. The study demonstrates the benefits of using high resolution airborne optical data to study in detail temporal and spatial variations of snow properties. It offers a new insight for the study of snow properties variability avoiding the limitations of point measurements. I think however that several questions should be addressed before it can be published.

**Main Comments**

1/ My first main comment focused on the grain size retrieval accuracy in presence of liquid water. It is stated section 3.3 that the range of wavelengths used in the grain size retrieval is 1.17 to 1.27 µm allowing to reduce the "bias (induced) by liquid water" with reference to Painter et al., 2013. In Painter et al., 2013, it is on the contrary stated that the retrieval is done using 1.03 to 1.06 µm wavelengths (section 4.3) and that the "technique could be less sensitive to biases due to liquid water and water vapor". It is also stated that this assumption is not based on field measurements. Are the range of wavelengths used similar in this study and in Painter et al., 2013 or not ? In my mind, the assumption concerning the limited impact of liquid water on the retrieval should be discussed and demonstrated in more depth.

**We discuss this now in a bit more depth. See our reply to reviewer #1. Our retrievals technique, as well as the contact probe-based methods in the field, can use both, the 1.27 or the 1.03 micron ice absorption feature.**
* * *
For example, considering figure 1 in Gallet et al., 2014a or Green et al., 2002, there are some differences between the ice and water refractive indices in the considered range of wavelengths. This can induce some errors on the grain size retrieval and should probably be included in the discussion. The algorithms developed in Green et al., 2002 can also probably be used to investigate in more detail the effect of liquid water of the grain size retrieval and also to identify on the AVIRIS data snow with liquid water. I think this point is really crucial to infer the uncertainty of grain size retrieval and to strengthen the discussion on the grain size spatial and temporal variations presented in this study.

*Gallet, J.-C., Domine, F., and Dumont, M.: Measuring the specific surface area of wet snow using 1310 nm reflectance, The Cryosphere, 8, 1139-1148, doi:10.5194/tc-8- 1139-2014, 2014a.*

*Green, R. O., Dozier, J., Roberts, D., Painter, T. (2002). Spectral snow-reflectance models for grain-size and liquid-water fraction in melting snow for the solar-reflected spectrum. Annals of Glaciology, 34(1), 71-73.*

**Please see Fig. R1 below showing the spectral absorption of the three water phases. The use the spectral range in our method is a compromise between using enough bands (enough signal) vs staying away from information coming from liquid or gaseous water. We completely agree with Dr. Dumont's comment, but we believe that this topic would deserve its own full paper. For this paper, we wanted to focus on the application of the described methods and specifically on the related results. We try to maintain this balance and would argue to postpone a more detailed discussion of the snow grain size retrieval algorithm to another dedicated paper. Moreover, all our experience using this algorithm in different spectral ranges with different band and even different sensors indicate that the result remain qualitatively the same, while the absolute retrieved microns chance. However, the absolute numbers in snow grain size are less relevant this paper in respect to their relative distribution space and change over time.**

**Note that we already refer to Gallet et al, 2014 on page 7, line 15.**

[Figure]

**Figure R1. Refractive index (log scale) wrt wavelength of the three phases of water. The legend reports the data sources.**
* * *
2/ The second main comment focuses on the grain size/elevation inverse relationship in case of large impurity load discussed in section 5. I think this is a really interesting finding and it would result in a negative feedback. It also has to be linked to other recent studies showing that light-absorbing impurity can reduce the density of melting snow (Meinander et al., 2014) or studying the effect of soot on anisotropy factor (Peltoniemi et al., 2015) However, the discussion should be strengthen to be more convincing. The first thing is that the mean decrease of grain size is roughly 50 μm (figures 7 and 9) which is close to the accuracy of the grain size retrieval method (the one given in Painter et al., 2013). Consequently, I think before discussing the possible physical causes of the grain size/elevation inverse relationship, the authors should probably discuss the significance of the grain size decrease relatively to differences sources of errors. There are several sources of errors that might have to be taken into consideration e.g. : a/ grain size retrieval accuracy in presence of liquid water (see my previous comment) b/ the fact that as stated by the authors the surface roughness is high. Consequently the HDRF derived from spherical model are probably more anisotropic that the natural surface inducing a larger error on the grain size for the rough surface than for flat one. c/ Peltoniemi et al., 2015 (see full reference below) also demonstrated that under large impurity load, the impurity sank in the snow and largely modify the HDRF . . . this also leads to an error in the grain size retrieval. d/ Finally, as discussed in Skiles PhD thesis (e.g. chapter 3, figure 1), it is possible that for really high impurity content, the snow reflectance is more dust-like than snow-like in the IR (and then higher in the NIR not due to snow grain size but to impurity particles). This effect seems to be overestimated by SNICAR (compared to field measurements) but it can still affect the grain size retrieval and results artificially smaller grain size.

Lastly, the processes leading to a decrease of snow grain size could probably further discussed : the presence of a large content of impurity largely modifies (and shortens) the light penetration depth so that the energy is absorbed closer to the surface. The temperature profile and the heat and vapour transfer are consequently modified leading to modified metamorphism. Note that the formation of sublimation crystals has also been discussed in Antarctica (Gallet et al., 2014b, see full reference below).

*Meinander, O., Kontu, A., Virkkula, A., Arola, A., Backman, L., Dagsson- Waldhauserová, P., Järvinen, O., Manninen, T., Svensson, J., de Leeuw, G., and Lep- päranta, M.: Brief communication: Light-absorbing impurities can reduce the density of melting snow, The Cryosphere, 8, 991-995, doi:10.5194/tc-8-991-2014, 2014.*

*Peltoniemi, J. I., Gritsevich, M., Hakala, T., Dagsson-Waldhauserová, P., Arnalds, Ó., Anttila, K., Hannula, H.-R., Kivekäs, N., Lihavainen, H., Meinander, O., Svens- son, J., Virkkula, A., and de Leeuw, G.: Soot on Snow experiment: bidirectional re- flectance factor measurements of contaminated snow, The Cryosphere, 9, 2323-2337, doi:10.5194/tc-9-2323-2015, 2015.*

*Gallet, J.-C., Domine, F., Savarino, J., Dumont, M., and Brun, E.: The growth of sub- limation crystals and surface hoar on the Antarctic plateau, The Cryosphere, 8, 1205- 1215, doi:10.5194/tc-8-1205-2014, 2014b.*

**It is true that absolute retrieval accuracy of snow grain size might be in the order**

of the changes observed in late June 2011, however, we believe that the observed trend is due to physical processes and not retrieval error. In other words, we are confident in tracing trends in snow grain sizes across a single image and from observation to observation.  Especially given that the inverse relationship with elevation and grain size has been observed in imagery from other years, and a decrease in optical grain size was observed in the field in presence of heavy dust loading/absence of new snowfall.

We would like to note that the surface roughness mentioned in the paper was observed by co-author Skiles in the field as the upper layers of snowpack transitioned from relatively clean to dirty snow over the coarse of a few days in 2013.  High resolution ground observations like these were not available for 2011 when the over fights presented here were taking place, and therefore we cannot say whether or not there was surface roughness during time of acquisition. We do recognize that surface roughness can affect the bidirectional reflectance distribution function of snow. That being said, we note here (from Painter et al., 2003) 'that Warren et al. (1998) found the effect of sastrugi negligible for near-nadir views. Moreover, they found that sastrugi have little effect on the BRDF for view angles less than 30° in the principal plane and for view zenith angles less than 50° for $\lambda$=0.9 µm and solar zenith angle $\theta_0$=67°, an effect that should be even smaller for smaller zenith angles.' Therefore, the assumption that surface roughness has a negligible impact on snow bidirectional reflectance should be valid for the AVIRIS scenes examined here.

We agree that this is an interesting finding, and deserves attention, but believe a more extended discussion on the processes leading to a decrease of snow grain size is outside the scope of this paper, which focuses on ability to monitor snow cover trends using imaging spectroscopy. There are two papers that are currently in preparation by co-author Skiles, based on her dissertation work, that address this process in much more detail, including the field observations and assessing the impact of increasing dust concentrations/water content on NIR absorption and grain size retrieval.

We address this in the manuscript by modifying the second sentence in the paragraph that discusses the grainsize inversion such that this paragraph becomes: "*Second, when there is an actual inversion of grain sizes with elevation, we suggest this is accounted for by rapid melt induced by high impurity content at the surface. Although the change in grain size is on the order of the uncertainty in grain size retrieval, it is not the absolute values that are important but rather the trend.  A decrease in surface grainsize in the presence of high impurity content was also observed in the daily field observations in 2013 by Skiles (2014), when the dust layers coalesced at the surface and skies were clear both grain sizes and density in the surface layers decreased and surface roughness increased.*"
* * *
3/ The third main comment focuses on the way the data are presented in Figures 8 and 9 with respect and elevation. I think the figures really nicely described the variations of the snow parameters with respect to topographic parameters. However, especially at the end of the season, when the snowpack remains mainly on shaded slopes this way to present the data could induce a bias in the interpretation. In other words, I mean that it is, for example, not straightforward from Figure 9, 3rd row that the inverse grain/size elevation relation ship is not a bias due to uneven distribution of snow cover with elevation and aspect. Consequently, my suggestion would be to plot the value of mean grain size (colour) as a function of aspect (x-axis) and elevation (y-axis) with a representation of the RMSE or the number of points in each topographic class (transparency or size of the dots).

**The sun is rising much higher (decreasing solar zenith angles) later in the ablation period where we observe the inverse relationship between grain size and elevation. In June, there are almost no shadows left at the time of observation, which typically is close to the local solar noon. We therefore do not think the shadows would introduce a significant bias or even have an impact on our conclusions.**

**We appreciate the suggested plot, which would bring the elevation and the terrain dependence into one single panel. We experimented extensively with this and other types of plots and we found the current plot to be most effective in illustrating the dependencies. The main reason for that is that the current plots show the density or number of occurrences by using 2D histograms. This density carries additional information, which is very helpful in the visual interpretation of the results. With the suggested scatterplot, however, the reader has to imagine the clustering, which is not easy in the ablation cases where we see a low dependency of snow grain size with respect to the terrain (see Fig. R2). We also created polar plots to simplify the interpretation of the terrain aspect, but we were also not happy with this. After careful consideration, we therefore believe that our current plots are optimal for the abovementioned reasons.**

[Figure]

**Figure R1. Experimental plots addressing Dr. Dumont's suggestion (3/). North it up in the polar plots and the elevation is plotted as radius.**
* * *
4/ The last main comment focuses on the uncertainties analysis provided in the paper. Though I perfectly agree that uncertainty analysis is not the main aim of the paper, I think that a deeper analysis of the errors could help to strengthen the conclusions of the paper as I explained in the previous comments. The authors refers to the uncertainties of the retrieval method provided in Painter et al., 2013 as being 25um for grain size, 0.0001 (there must be a typo there) for snow albedo and 4 Wm2 for radiative forcing. In Painter et al. 2013, the uncertainties given for albedo is 0.001-0.004 (based on two measurements only. . .), 2.1 +-5.1 Wm2 for radiative forcing, and the uncertainty of the grain size retrieval is an assumption. In Painter et al., 2013, only one date was available for AVIRIS data but with all the dates used in the study, I am wondering if the uncertainty data could used more broadband albedo measurements to strengthen the uncertainty analysis. Section 4.5. presents really interesting results regarding the comparison of two images acquired only one day apart from each other but it would worth a more detailed analysis. Are the given values mean values ? What are the spreads of the difference ? In my mind, the uncertainty analysis should be more careful, describing which values are assumed, what is the significance of the different values (how many measurements are used to derive the uncertainty). . .

**We realize that having two sections with error/accuracy discussions might be confusing. We therefore decided to merge the two sections into Sect. 4.5. We have completely rewritten this section and we now account and correct for all of the points raised by this reviewer's comment.**
* * *
**Specific Comments**

Page 2, line 13 : Maybe the authors can be a bit more explicit why the SSA /optical grain radius relationship is not a strict equality.

**We believe that the this is covered by the two references and further discussion is out of scope for this paper.**
* * *
Section 3.3 . The modeled HDRF must account for the diffuse to total incoming irradiance ratio. Therefore it could worth explaining how this ratio is calculated. For this section, please see also my main comment 1/.

**The ratio is calculated from diffuse and total incoming irradiance, which are both produced by ATCOR-4 (the authors specifically requested this capability to simplify automated processing of imaging spectrometer products from the Airborne Snow Observatory). We add language to specifically state this in Sect. 3.4 where we introduce the (total) irradiance term.**
* * *
Page 7, Equation 7 : If c is meant to account for imperfect terrain corrections, c probably also have a second order dependency to wavelength. This could be maybe detailed in the paper.

**Yes, c is some sort of calibration to remove imperfections, mainly induced by the terrain. We reason why we choose 1.08 microns is already described on page 7, line 4 and 5.**
* * *
Page 7, line 13 : *This assumption becomes more valid with grain growth associated with evolving metamorphism and rounding of the snow crystals*. Rounded forms do not necessarily means perfect spheres, and spheroids optical properties might significantly differ from spheres optical properties (e.g. Libois et al., 2013, full ref below, fig1). The differences between spheres and spheroids optical properties might even be larger that the ones between spheres and other snow crystals optical properties. Consequently, in my mind, this sentence should be either removed or further justified.

*Libois, Q., Picard, G., France, J. L., Arnaud, L., Dumont, M., Carmagnola, C. M., and King, M. D.: Influence of grain shape on light penetration in snow, The Cryosphere, 7, 1803-1818, doi:10.5194/tc-7-1803-2013, 2013.*

**We removed this sentence.**
* * *
Page 8, lines 9-13 : The mean broadband albedo varies not only with the snow properties but also with the illumination geometry and consequently the date and time of acquisition. This should probably be discussed while presenting the results of Figure 7.

**Albedo is higher when the sun in closer to the horizon, but the difference is small relative to impact of grain size/impurity content. We added the following text to the first paragraph of Sect. 4.5: "*While inherent snow albedo is controlled by grain size and impurity content over the course of a day, the solar zenith angle, cloud cover, and topography also act to influence albedo and net solar radiation (Warren, 1982). For our acquisitions, we account for local illumination and view geometries in the albedo retrievals (Eq. 2 and 5).*"**
* * *
Section 4.3. Same comment as above. Snow albedo depends not only on the snow properties but also on the illumination conditions. This has, in my mind, to be taken into account in the discussion regarding the spatial variations of albedo.

**Local illumination and view geometries are accounted for in our algorithms (See Eq. 2).**
* * *
**Minor Comments**

Page 2, line 16 : Changes is surface roughness also cause changes in snow albedo

**We added surface roughness.**
* * *
Page 4, line 22 : "retrieved columnar water vapor", maybe the authors could explain how, and from which data and add a reference.

**We mention ATCOR-4 here and added the corresponding reference.**
* * *
Page 8, line 18 : Surface snow grain size can also be affected by wind effects.

**We added the wind effects.**
* * *

[revised manuscript text omitted]
 Flight Date | Scene Nr. | Aircraft Altitude [km] | Heading [deg.] | Mean Elevation [km] | Pixel Size [m] | Solar Zenith [deg.] | Azimuth [deg.] | Water Vapor [cm] |
|---|---|---|---|---|---|---|---|---|
| Sierra Nevada, CA. 2009 | | | | | | | | |
| 27 Feb | 6 | 19.799 | 354 | 2.547 | 14.1 | 48.3 | 205.7 | 0.32 |
| | 8 | 19.836 | 185 | 2.925 | 13.9 | 56.1 | 205.4 | 0.23 |
| 05 Mar | 5 | 19.806 | 181 | 2.056 | 14.4 | 45.7 | 155.3 | 0.22 |
| | 6 | 19.765 | 000 | 2.554 | 14.0 | 44.7 | 158.7 | 0.13 |
| | 7 | 19.773 | 181 | 2.879 | 13.7 | 44.1 | 162.7 | 0.09 |
| 04 Apr | 5 | 19.799 | 182 | 2.082 | 14.4 | 41.0 | 131.6 | 0.16 |
| | 6 | 19.674 | 359 | 2.552 | 13.9 | 39.5 | 134.4 | 0.10 |
| | 7 | 19.902 | 181 | 2.911 | 13.8 | 38.5 | 137.3 | 0.07 |
| | 8 | 19.796 | 000 | 1.865 | 14.5 | 37.4 | 139.7 | 0.24 |
| 07 May | 6 | 14.262 | 348 | 3.020 | 9.0 | 23.4 | 144.0 | 0.30 |
| | 7 | 14.274 | 193 | 2.650 | 9.3 | 22.4 | 148.6 | 0.43 |
| | 8 | 14.273 | 349 | 2.277 | 9.3 | 21.5 | 154.4 | 0.62 |
| 08 May | 5 | 14.141 | 343 | 3.178 | 8.9 | 27.6 | 129.0 | 0.36 |
| | 6 | 14.154 | 194 | 2.642 | 9.2 | 26.4 | 132.2 | 0.55 |
| | 7 | 14.154 | 345 | 2.383 | 9.2 | 25.0 | 136.4 | 0.73 |
| | 8 | 14.144 | 195 | 2.035 | 9.6 | 23.9 | 140.7 | 0.89 |
| | 9 | 14.157 | 346 | 2.715 | 9.2 | 22.6 | 146.1 | 0.58 |
| Rocky Mountains, CO. 2011 | | | | | | | | |
| 13 May | 5 | 6.496 | 352 | 3.560 | 2.1 | 19.6 | 173.0 | 0.25 |
| | 6 | 6.536 | 354 | 3.526 | 2.1 | 19.5 | 182.9 | 0.25 |
| | 7 | 6.533 | 354 | 3.683 | 2.1 | 19.9 | 192.6 | 0.21 |
| 16 May | 7 | 19.866 | 180 | 3.237 | 13.5 | 20.5 | 153.5 | 0.22 |
| 09 Jun | 12 | 20.170 | 179 | 3.260 | 13.7 | 15.0 | 179.1 | 0.33 |
| 15 Jun | 12 | 20.216 | 180 | 3.196 | 13.8 | 20.8 | 129.7 | 0.51 |
| 23 Jun | 5 | 20.139 | 001 | 3.271 | 13.7 | 38.3 | 102.0 | 0.54 |

[Figure]

**Figure 1.** Geographic extend of the AVIRIS image data. The study area is shown hatched. It is based on a mosaic of subsets, such that a common area exists for all flight dates. The elevation model is hill shaded in the background with an illumination corresponding to the mean solar position for all dates.

[Figure]

**Figure 2.** Histograms of the terrain aspect (top) and elevation (bottom) for the two study sites, Kings/Kaweah Basin Sierra Nevada, CA. (left) and Uncompahgre Basin Rocky Mountains, CO. (right). The values are normalized by the size of the study area given in Table 1. The peak occurrences of terrain aspect give an indication to the main drainage direction of a river basin.

[Figure]

**Figure 3.** Snow accumulation (blue) and snowmelt (red) indicated with the differential Snow Water Equivalent ($\Delta$ SWE) with respect to time. The AVIRIS flight dates are marked with black vertical lines and a symbol indicating the ER-2 or Twin-Otter airborne platform. The upper plot shows snow sensor pillow data from two stations representing a higher and a lower location in the Sierra Nevada, CA, study area in 2009. The lower plot shows the same information, but from stations close to the Rocky Mountains, CO, study area in 2011.

[Figure]

**Figure 4.** Maps of snow grain size retrieved using Eq. (4) for three dates during the melt period in the Sierra Nevada (upper panel), and the Rocky Mountains (lower panel). The upper and lower map represent 252 km$^2$ and 55 km$^2$, respectively (Table. 1).

[Figure]

**Figure 5.** Same as Fig. 4 but for snow albedo.

[Figure]

**Figure 6.** Same as Fig. 4 but for snow radiative forcing.

[Figure]

**Figure 7.** Time-series of snow cover, albedo, grain size and impurity radiative forcing in the Sierra Nevada, CA., USA in 2009 (upper panel), and in the Rocky Mountains, Co., USA in 2011 (lower panel). The snow cover is normalized by the size of the study area and the other values represent the average value per date surrounded by the corresponding standard deviation.

[Figure]

**Figure 8.**  Two-dimensional histogram of snow grain size with respect to terrain elevation (left) and aspect (right) in the Sierra Nevada in spring 2009.

[Figure]

**Figure 9.** Same as Fig. 8 but for the Rocky Mountains in spring 2011. Note: A rare retrieval artifact is found on 15 June where Eq. 4 has no unique solution within a snow grain size range of 540 to 570 $\mu$m causing a gap in the corresponding plots.

[Figure]

**Figure 10.** Snow albedo along a North–South (left), and West–East transect (right) through the middle of the observed Sierra Nevada area. The values are smoothed with a boxcar average of 30 pixels. The snow-free surface is shown in black.

[Figure]

**Figure 11.** Same as Fig. 10 but for the Rocky Mountains.

[Figure]

**Figure 12.** Snow albedo with respect to terrain elevation (left) and terrain aspect (right) in the Sierra Nevada in spring 2009.

[Figure]

**Figure 13.** Same as Fig. 12 but for the Rocky Mountains in spring 2011.

[Figure]

**Figure 14.** Radiative forcing of snow impurities with respect to terrain elevation (left) and aspect (right) in the Sierra Nevada in spring 2009.

[Figure]

**Figure 15.** Same as Fig. 14 but for the Rocky Mountains in spring 2011.

---

## Author Response (AR2)

**TC-2015-196**

**Author Response to Editor's Comment**

We are very grateful for the thoughtful comments by the editor. They helped considerably to improve our manuscript.
Below, please find the editor's comments in serif font and our **replies in bold sanserif font**.
We use a ***bold sanserif italic font*** to identify new or changed text.
We applied few minor improvements to enhance the readability of the manuscript.
Please find all changes highlighted ( and new) in the appended manuscript.

Editor Decision: Publish subject to minor revisions (Editor review) (18 Apr 2016) by Florent Dominé
Comments to the Author:
Dear Authors

Thanks you for your revised version of manuscript tc 2015-196. Your new version is significantly improved. However, I contacted the referees with specific questions and they indicated that several of your changes did not quite respond to their objections.

In general, I do find your paper of high scientific quality, of great originality, and I will be happy to accept it for publication in The Cryosphere. However, it is probably in your interest, in the interest of the journal and in the interest of our wide readership that some aspects be clearer or explained with a bit more detail and with more rigor. I will just focus here on the point that caused significant dissatisfaction with the reviewer.

Your treatment of uncertainties appears unsatisfactory and your response to point 2 by Dr. Dumont does not really address the elements she raised. Your response is essentially "We believe…", and I get the feeling that the main effect of your changes is to sweep the difficulty under the rug rather than actually try to improve your text. I very strongly recommend that you address the uncertainty issue in this paper, even though subsequent papers may also add further detail. Furthermore, the observed decrease in grain size is not really addressed, while the growth of sublimation crystals may be a possible interesting hypothesis. Their formation may be favored by the accumulation of impurities near the surface which will modify the profile of energy absorption and of temperature. While I understand that you may not want to significantly expand your paper on that subject, throwing in a few interesting ideas or suggestions, possibly developed in subsequent papers, would be in order.

Regarding a simple request by referee 1, it would probably be still useful to mention clearly in the introduction that the radiative forcing referred to is always by LAISI. Sometimes you mention "radiative forcing by LAISI" and sometimes just "radiative forcing", but you never explicitly equate both terms. This may be ambiguous to non-specialists and I therefore recommend that you state clearly that when you mention radiative forcing, it is by LAISI.

Addressing these issues should not take too much time. I will be happy to accept your paper once you have addressed appropriately these simple requests.

Best regards,

Florent Domine

**Dear Florent Domine**

**Thank you for your valuable comments.**

**We now discuss in detail the uncertainties in the grain size retrieval in the presence of liquid water and presence of impurities. This discussion points to previous literature but also points to directions forward for investigation, given the importance of the cryosphere energy balance, melt rates, and snow-albedo feedback.**

**We now added a sentence to the introduction to clarify that we refer to "radiative forcing by LAISI" at all times.**

**We hope that this addresses all of your comments.**

**Thank you,**
**Felix Seidel**

[revised manuscript text omitted]